# Model soups need only one ingredient

**Alireza Abdollahpoorrostam** [EPFL] **Nikolaos Dimitriadis** [EPFL] **Adam Hazimeh** [EPFL] **Pascal Frossard** [EPFL]

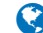

## Abstract

Fine-tuning large pre-trained models on a target distribution often improves in-distribution (ID) accuracy, but at the cost of out-of-distribution (OOD) robustness as representations specialize to the fine-tuning data. Weight-space ensembling methods, such as Model Soups, mitigate this effect by averaging multiple checkpoints, but they are computationally prohibitive, requiring the training and storage of dozens of fine-tuned models. In this paper, we introduce MonoSoup, a simple, data-free, hyperparameter-free, post-hoc method that achieves a strong ID–OOD balance using *only a single* checkpoint. Our method applies Singular Value Decomposition (SVD) to each layer's update and decomposes it into high-energy directions that capture task-specific adaptation and low-energy directions that introduce noise but may still encode residual signals useful for robustness. MonoSoup then uses entropy-based effective rank to automatically reweight these components with layer-wise coefficients that account for the spectral and geometric structure of the model. Experiments on CLIP models fine-tuned on ImageNet and evaluated under natural distribution shifts, as well as on Qwen language models tested on mathematical reasoning and multiple-choice benchmarks, show that this plug-and-play approach is a practical and effective alternative to multi-checkpoint methods, retaining much of their benefits without their computational overhead.

## 1. Introduction

The *pre-train-then-finetune* paradigm (Kumar et al., 2022) has become the de-facto approach for leveraging the capa-

[EPFL]LTS4, École Polytechnique Fédérale de Lausanne (EPFL), Switzerland. Correspondence to: Alireza Abdollahpoorrostam <alireza.abdollahpoorrostam@epfl.ch>.

*Proceedings of the $43^{rd}$ International Conference on Machine Learning*, Seoul, South Korea. PMLR 306, 2026. Copyright 2026 by the author(s).

bilities of foundation models (Bommasani et al., 2022) and has accelerated progress across a wide range of applications (Radford et al., 2021; Rombach et al., 2022). However, specialization often comes at a cost: the fine-tuning process that adapts a model to a target distribution frequently degrades its general-purpose knowledge, leading to a significant drop in out-of-distribution (OOD) performance, a phenomenon known as *catastrophic forgetting* (McCloskey & Cohen, 1989). This leads to a trade-off between in-distribution (ID) performance and OOD robustness, which remains a central challenge for the reliable deployment of these powerful models (Kumar et al., 2022; Goyal et al., 2023).

To address this trade-off, post-hoc methods that directly manipulate model weights have gained traction. A prominent example is Model Soups (Wortsman et al., 2022a), which improves both ID and OOD performance by averaging the weights of multiple fine-tuned checkpoints. While effective, this approach is often impractical due to the computational and storage overhead of training and retaining dozens of checkpoints. To reduce this burden, ModelStock (Jang et al., 2024) was proposed as a more efficient alternative, requiring only two models and weighting their updates according to their geometric alignment. However, this assumption of having access to two suitable checkpoints is often unrealistic in practice, as model repositories typically store only a single, best-performing version. In parallel, Wise-FT (Wortsman et al., 2022b) explored single-model robustness by interpolating between the pre-trained and fine-tuned weights, leveraging the low-loss path between them. While effective in tracing the trade-off between specialization and robustness, Wise-FT applies a uniform interpolation across all layers and directions, leaving fine grained anisotropic effects unaddressed. These observations naturally raise the following question:

> *"Can we retain the benefits of model soups when only a single fine-tuned model is available?"*

In this paper, we answer this question affirmatively. We begin by characterizing when multi-model merging succeeds and when it fails. Our analysis reveals that improvements in both ID and OOD accuracy are consistently obtained when the fine-tuning updates of two models are well-aligned (high

cosine similarity), highlighting a strong link between geometric similarity and generalization. To validate this insight, we introduce a Similarity-Filtered Greedy Soup that selects only those models that would improve the overall geometric alignment of the soup. This simple variant both confirms our hypothesis and provides a data-free, computationally efficient alternative to standard soups. These findings align with a unifying principle from recent works (Wortsman et al., 2022a; Jang et al., 2024; Ramé et al., 2023; Gargiulo et al., 2025; Stoica et al., 2025) that successful weight-space ensembling methods reinforce dominant directions that encode task-relevant signals, while suppressing noisy or misaligned directions that degrade both ID and OOD performance.

Building on these insights, we shift from analyzing pairs of models to studying the internal properties of a single fine-tuned checkpoint. Since such models often over-specialize at the cost of OOD robustness, our goal is to test whether the structural signals that enable successful merging across multiple models can be exploited within a single model's weight space. To this end, we propose MonoSoup, a data-free and hyperparameter-free approach that applies Singular Value Decomposition (SVD) to each layer's update and decomposes it into two complementary components: a principal subspace capturing high-energy directions associated with task-specific knowledge, and an orthogonal complement capturing low-energy directions that preserve information critical for OOD generalization. MonoSoup reweights these components using principled, layer-wise coefficients that automatically adapt to the model's spectral and geometric properties. Crucially, MonoSoup leverages the entropy-based effective rank (Roy & Vetterli, 2007) to automatically partition each layer's weight update, enabling robust adaptation to heterogeneous layer dynamics and architectures, including both Transformers and CNNs, without manual tuning. The resulting single edited checkpoint achieves a better balance between specialization and generalization.

Extensive experiments show that MonoSoup matches or exceeds the performance of multi-model methods while using just one fine-tuned checkpoint. On CLIP (Radford et al., 2021) models fine-tuned on ImageNet (Deng et al., 2009), it improves the average OOD accuracy of the strongest baseline by $\sim 1\%$ and recovers up to 8% on weaker, representation-collapsed checkpoints, while maintaining strong ID accuracy. Similar gains are also observed on language-based mathematical reasoning and QA tasks using Qwen3 (Yang et al., 2025). Moreover, MonoSoup complements single-model techniques such as Wise-FT (Wortsman et al., 2022b), providing a stronger checkpoint that further improves the ID–OOD trade-off when the two are combined.

Our contributions are the following:

1. We establish a geometric perspective on when model merging succeeds or fails, showing that performance

is closely tied to the alignment of fine-tuning updates. This analysis clarifies principles underlying multi-model methods and motivates their extension to the single-checkpoint setting.

2. Based on this analysis, we test a new baseline method, Similarity-Filtered Greedy Soup, which is a data-free variant of the original method that uses geometric alignment as a selection criterion. This baseline demonstrates that alignment is a reliable proxy for merging effectiveness.

3. We then introduce MonoSoup, a data-free, hyperparameter-free, post-hoc editing approach that improves the ID-OOD trade-off using only a single fine-tuned model, which typically suffers from degraded OOD performance on its own due to representation collapse. Our method decomposes each layer's update into high- and low-energy components and adaptively reweights them, eliminating the need for multiple checkpoints.

4. We empirically validate our approach on vision (CLIP, ConvNeXt) and language (Qwen) benchmarks, demonstrating that MonoSoup consistently improves OOD generalization while maintaining or enhancing in-distribution accuracy.

## 2. Preliminaries

**Model Soups.** The common practice in machine learning is to select the single best model from a hyperparameter search for final deployment, based on a validation metric, and discard the remaining checkpoints. However, models originating from the same pre-trained initialization often occupy a connected, low-loss basin in the optimization landscape (Garipov et al., 2018; Frankle et al., 2020; Izmailov et al., 2018), making them amenable to ensembling. Formally, consider $m$ weights $\{\boldsymbol{\theta}_t\}_{t \in [m]}$, obtained by fine-tuning the pre-trained weights $\boldsymbol{\theta}_0$ on a target dataset $\mathcal{D}_{\text{train}}$ with $m$ different hyperparameter configurations. Model Soups (Wortsman et al., 2022a) leverages this insight by averaging the weights of multiple fine-tuned models, resulting in the final parameters $\boldsymbol{\theta} = \frac{1}{T}\sum_{t=1}^{T}\boldsymbol{\theta}_t$. While effective, soups are computationally expensive: their benefits are most pronounced when averaging many diverse checkpoints (Ramé et al., 2023; 2022), which is impractical for large-scale models.

**Model Stock.** In an effort to reduce the significant computational and storage overhead of Model Soups, Model Stock (Jang et al., 2024) requires only two models and operates layer-wise, based on the idea that cosine similarity can quantify the signal-to-noise ratio of the fine-tuning updates. Let $\boldsymbol{W}_0^{(\ell)}$ denote the pre-trained weights at layer $\ell$, and $\boldsymbol{W}_1^{(\ell)}, \boldsymbol{W}_2^{(\ell)}$ the corresponding updates from the two checkpoints. Model Stock first computes the cosine simi-

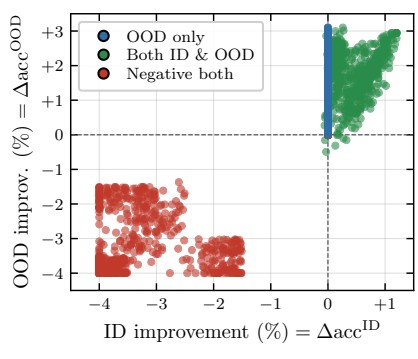

*(a)* Performance scatter.

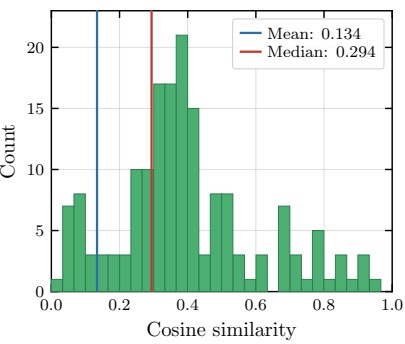

*(b)* Low-performing pair similarity.

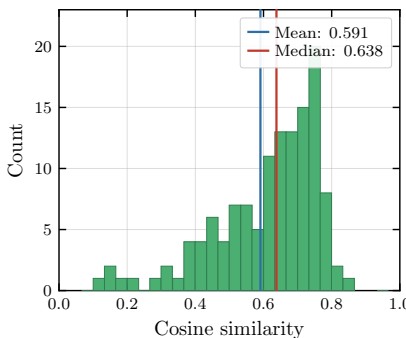

*(c)* High-performing pair similarity.

*Figure 1.* Performance and alignment analysis of Model Stock on 2,409 pairwise combinations of CLIP ViT-B/32 models fine-tuned on ImageNet. (a) Scatter plot of ID vs. OOD performance relative to the better constituent model. (b) and (c): Layer-wise cosine similarity for low-performing and high-performing, respectively. Stronger alignment coincides with consistent gains, highlighting that alignment can serve as a key predictor of merging success.

larity $\cos \alpha^{(\ell)}$ between these task vectors to measure their agreement. The merged weights are:

$$\boldsymbol{W}_{\text{stock}}^{(\ell)} = \boldsymbol{W}_0^{(\ell)} + \lambda^{(\ell)} \cdot \left( \frac{\boldsymbol{W}_1^{(\ell)} + \boldsymbol{W}_2^{(\ell)}}{2} \right), \quad (1)$$

where the scaling factor is defined as $\lambda^{(\ell)} = \frac{2\cos\alpha^{(\ell)}}{1+\cos\alpha^{(\ell)}}$. This rule preserves directions where the updates are well-aligned ($\cos\alpha^{(\ell)} \to 1$), while reverting toward the pre-trained weights when they disagree. Compared to soups, it is more efficient since it reduces the number of required models from a large ensemble to just two, but still assumes access to at least two fine-tuned checkpoints.

**Wise-FT and LiNeS.** Beyond multi-checkpoint methods, there also exist approaches that operate with a single fine-tuned model. Wise-FT (Wortsman et al., 2022b) improves generalization by linearly interpolating between the pre-trained and fine-tuned weights. Given a coefficient $\lambda \in [0, 1]$, the merged parameters are $\boldsymbol{\theta}_{\text{wise}} = (1-\lambda)\boldsymbol{\theta}_0 + \lambda\boldsymbol{\theta}_t$, which traces a continuum between robustness (closer to $\boldsymbol{\theta}_0$) and specialization (closer to $\boldsymbol{\theta}_t$). Therefore, Wise-FT delivers a family of models along this path rather than a single edited checkpoint, and it applies the same interpolation uniformly across all layers and directions, limiting its ability to capture anisotropic updates. LiNeS (Wang et al., 2025a) also operates on a single checkpoint, introducing post-training layer scaling to prevent catastrophic forgetting (McCloskey & Cohen, 1989) and enhance model merging. However, it requires labeled data to tune its hyperparameters and employs a single coefficient for all layers within a transformer block, potentially overlooking the distinct dynamics of linear and attention layers.

## 3. The role of alignment in model merging

In this section, we investigate the conditions under which multi-model merging succeeds, finding that success often

depends on the alignment of the fine-tuning updates. To investigate this, we use Model Stock (Jang et al., 2024) as a probe: since its layer-wise weighting rule explicitly depends on cosine similarity as in Equation 1, it provides a natural lens to study how alignment relates to merging performance.

We evaluate all pairwise combinations among 70 CLIP ViT-B/32 models, released by Wortsman et al. (2022a) and fine-tuned on ImageNet; each model corresponds to a different hyper-parameter configuration. We compare each merged model against the better of its two constituents across five natural distribution shifts: ImageNet-V2 (Recht et al., 2019), ImageNet-R (Hendrycks et al., 2021a), ImageNet-Sketch (Wang et al., 2019), ImageNet-A (Hendrycks et al., 2021d), and ObjectNet (Barbu et al., 2019). Specifically, we define the performance differences of model stock w.r.t. the involved models:

$$\Delta\text{acc}^{\text{ID}} = \text{acc}_{\text{MS}}^{\text{ID}} - \max\left\{\text{acc}_1^{\text{ID}}, \text{acc}_2^{\text{ID}}\right\}$$
$$\Delta\text{acc}^{\text{OOD}} = \text{acc}_{\text{MS}}^{\text{OOD}} - \max\left\{\text{acc}_1^{\text{OOD}}, \text{acc}_2^{\text{OOD}}\right\}$$

As illustrated in Figure 1a, which spans all $\binom{70}{2} = 2415$ pairwise combinations, merging outcomes are highly sensitive to the choice of pair: a substantial fraction of combinations degrade both metrics, while simultaneous ID and OOD improvement is limited to a subset of well-aligned pairs[1]. We randomly select a pair from each mode, showing a histogram of per-layer cosine similarities for a low- and high-performing pair in Figure 1b and Figure 1c, respectively. We observe performance improvements when task vectors are well aligned, but merging benefits diminish when they are weakly aligned. This sensitivity highlights a key principle: merging is effective when the fine-tuning updates are well-aligned, but fails when conflicting updates interfere.

To validate this observation, we introduce *Similarity-Filtered Greedy Soup* (SFGS), a data-free variant of Greedy

---

[1]Six pairs with extreme negative outliers are excluded from the figure to preserve axis scale; their results are reported in Table 9.

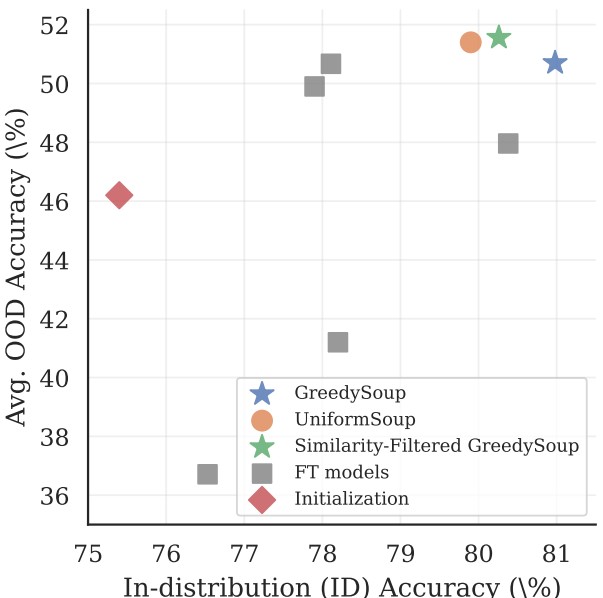

*Figure 2.* **Performance of Similarity-Filtered Greedy Soup (SFGS).** Evaluated on CLIP ViT-B/32 checkpoints, SFGS achieves competitive ID and OOD performance relative to validation-based greedy soup. This supports the finding that geometric alignment is a key indicator of merging effectiveness.

Soup, which replaces validation-based selection with a geometric filter. Starting with the best ID-performing model, we include a candidate checkpoint $\boldsymbol{\theta} = \boldsymbol{\theta}_0 + \boldsymbol{\tau}$, decomposed as the sum of the pre-trained weights $\boldsymbol{\theta}_0$ and the task vector $\boldsymbol{\tau}$, if its average layer-wise cosine similarity to the current soup exceeds a threshold $\delta$: $\frac{1}{|S|} \sum_{j \in S} \cos(\boldsymbol{\tau}, \boldsymbol{\tau}_j) \geq \delta$. As shown in Figure 2, this lightweight procedure, termed *Similarity-Filtered Greedy Soup*, achieves performance comparable to validation-based Greedy Soup (Wortsman et al., 2022a), showing that geometric alignment serves as a reliable proxy for effective merging. Nevertheless, like all soup-based methods, it still requires multiple fine-tuned checkpoints.

Taken together, these analyses suggest that successful merging of models originating from the same pre-trained initialization hinges on reinforcing well-aligned directions while suppressing noisy or conflicting ones. In the next section, we explore whether these principles apply within a single model: its fine-tuning update may contain both dominant, task-relevant directions as well as weaker components that can harm generalization.

## 4. MonoSoup

Our analysis of multi-model merging suggests that performance gains arise when fine-tuning updates reinforce shared directions while suppressing noisy or conflicting ones. This motivates searching for analogous signals *within a single checkpoint*. Specifically, we hypothesize that the update of a fine-tuned model contains both dominant

directions that capture task-specific adaptation and weaker components that, while less prominent, are important for maintaining generalization.

To make this structure explicit, we analyze the weight difference matrix at each layer $\boldsymbol{W}^{(\ell)} = \boldsymbol{W}_{\text{FT}}^{(\ell)} - \boldsymbol{W}_0^{(\ell)} \in \mathbb{R}^{m \times n}$, where $\boldsymbol{W}_0^{(\ell)}$ and $\boldsymbol{W}_{\text{FT}}^{(\ell)}$ are the pre-trained and fine-tuned weights for layer $\ell$, respectively. Applying singular value decomposition (SVD), $\boldsymbol{W}^{(\ell)} = \boldsymbol{U}^{(\ell)} \boldsymbol{\Sigma}^{(\ell)} \boldsymbol{V}^{(\ell)\top}$, where the spectrum of singular values $\sigma_1 \geq \sigma_2 \geq \dots$ quantifies how the adaptation is distributed across directions. We partition this spectrum into two components:

$$\boldsymbol{W}_{\text{High}}^{(\ell)} = \sum_{i \leq k} \sigma_i^{(\ell)} u_i^{(\ell)} v_i^{(\ell)\top}, \quad \boldsymbol{W}_{\text{Low}}^{(\ell)} = \boldsymbol{W}^{(\ell)} - \boldsymbol{W}_{\text{High}}^{(\ell)},$$

where the central design decision is the choice of the partition index $k$. We consider two approaches. The first fixes a spectral energy threshold $R \in [0, 1]$ and selects $k$ as the smallest index that captures at least a fraction $R$ of the total spectral energy:

$$k = \operatorname*{argmin}_{j} \left\{ j \;\middle|\; \frac{\sum_{i=1}^{j} \sigma_i^2}{\sum_{i=1}^{\min(m,n)} \sigma_i^2} \geq R \right\}. \qquad (2)$$

This variant is interpretable, but introduces the manual threshold $R$ as a hyperparameter. The second approach eliminates this requirement entirely by selecting $k$ automatically via the entropy-based effective rank (Roy & Vetterli, 2007):

$$k^{(\ell)} = \left\lceil \exp\left( -\sum_i p_i^{(\ell)} \ln p_i^{(\ell)} \right) \right\rceil, \qquad (3)$$

$$\text{where } p_i^{(\ell)} = \frac{\sigma_i^{(\ell)}}{\sum_j \sigma_j^{(\ell)}}. \qquad (4)$$

This adaptive boundary enables MonoSoup to dynamically accommodate the heterogeneous spectral properties across various layers and architectures, ensuring robust partitioning without manual hyperparameter selection. Intuitively, the high-energy spectral component $\boldsymbol{W}_{\text{High}}^{(\ell)}$ encodes concentrated task-specific adaptation, while the low-energy $\boldsymbol{W}_{\text{Low}}^{(\ell)}$ contains residual updates that, despite potentially capturing noise, may preserve information critical for OOD robustness.

Having established the partition, a natural question is whether $\boldsymbol{W}_{\text{Low}}^{(\ell)}$ can simply be discarded, that is, whether the low-energy component carries any signal worth preserving. Several recent studies (Gargiulo et al., 2025; Tang et al., 2025; Stoica et al., 2025) have argued that $\boldsymbol{W}_{\text{Low}}^{(\ell)}$ largely encodes noise and that discarding it can improve merging. These results, however, are mostly based on the standard task arithmetic benchmark (Ilharco et al., 2023), where the CLIP vision encoder is fine-tuned on a collection of relatively small-scale classification tasks, such as MNIST

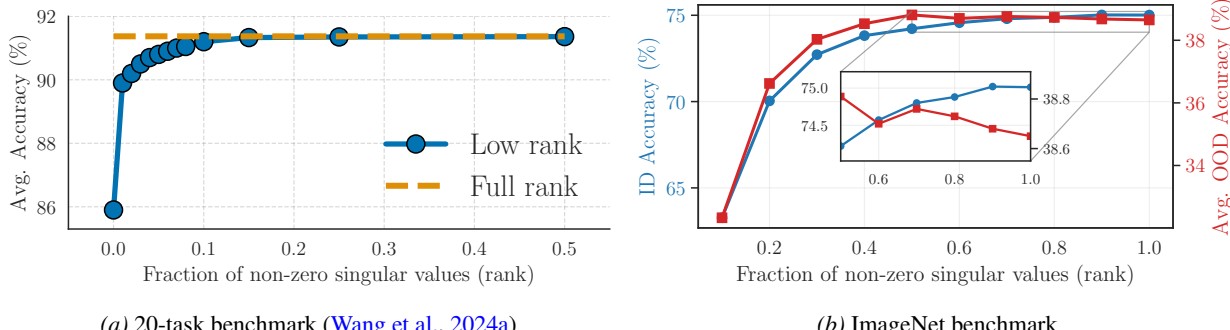

*(a)* 20-task benchmark (Wang et al., 2024a)       *(b)* ImageNet benchmark

*Figure 3.* Effect of truncating low-energy components on different benchmarks. (a) On the 20-task vision benchmark, performance saturates after retaining only a small number of singular values, consistent with prior reports that low-rank updates suffice. (b) On ImageNet with natural OOD shifts, truncation substantially reduces both ID and OOD accuracy, even when preserving 95% of spectral energy. This highlights that, in large-scale fine-tuning, low-energy directions carry critical information for generalization and cannot simply be removed. See Appendix D for further details.

(LeCun et al., 1998) and Cars (Krause et al., 2013). In Figure 3a, we progressively remove a larger amount of singular values and track the average performance on the suite of 20 tasks proposed by Wang et al. (2024a). In this regime, adaptation tends to be concentrated in a few dominant directions, so truncation appears effective.

In contrast, our setting involves fine-tuning on the significantly more complex ImageNet benchmark and evaluating across natural distribution shifts (ImageNetV2, ImageNet-R, ImageNet-Sketch, ObjectNet, ImageNet-A). As shown in Figure 3b, removing low-energy components in this regime leads to degradation of both ID and OOD accuracy, even when retaining 95% of spectral energy. This suggests that low-energy directions encode complementary information that is essential for OOD robustness.

To balance specialization and generalization, we reweight the high- and low-energy components rather than discarding one of them. For each layer, the edited update is

$$W_{\text{MonoSoup}}^{(\ell)} = \lambda_{\text{High}}^{(\ell)} W_{\text{High}}^{(\ell)} + \lambda_{\text{Low}}^{(\ell)} W_{\text{Low}}^{(\ell)}, \qquad (5)$$

where coefficients $\lambda_{\text{High}}^{(\ell)}$ and $\lambda_{\text{Low}}^{(\ell)}$ are determined adaptively. We now turn to how low-energy directions should be weighted relative to the dominant ones. We rely on two complementary signals derived directly from the model. The first comes from the singular value spectrum itself: when the decay is steep, most adaptation is captured by the leading singular values, suggesting that residual directions are less informative. When the singular value spectrum is flat, however, the contribution of weaker directions is more substantial. To capture this behavior, we define a spectral decay ratio,

$$\rho^{(\ell)} = \left( \frac{\sigma_{k+1}\left(W^{(\ell)}\right)}{\sigma_1\left(W^{(\ell)}\right)} \right)^2, \qquad (6)$$

which is small when the spectrum decays steeply and large when it is flat. The second signal represents the fractional

energy of the low-rank subspace:

$$\cos^2 \alpha^{(\ell)} = \frac{\left\| W_{\text{Low}}^{(\ell)} \right\|_F^2}{\left\| W^{(\ell)} \right\|_F^2} \in [0, 1] \qquad (7)$$

where $\cos \alpha^{(\ell)}$ measures the fraction of update energy carried by low-energy directions; see Appendix C for more details. Larger $\cos \alpha^{(\ell)}$ indicates that the fine-tuning update is spread over many weak directions rather than being concentrated in a small number of dominant singular vectors. As we show in Appendix G using CKA (Kornblith et al., 2019) on hidden representations, these low-energy directions preserve pretrained features on OOD inputs while providing only mild specialization on ID. While Figure 3 demonstrates that low-energy directions containing vital OOD signals may be present in the tail, Equation 7 ensures we *only* preserve this tail when the signal-to-noise ratio, captured by alignment $\cos \alpha$, warrants it, avoiding the noise injection of pure uniform averaging.

To determine the coefficients for $W_{\text{MonoSoup}}^{(\ell)}$ in Equation 5, we define a weighting rule that adapts to the model's spectral and geometric state. We require the low-energy coefficient $\lambda_{\text{Low}}^{(\ell)}$ to satisfy four natural boundary conditions: (i) Suppression: $\lambda_{\text{Low}} \to 0$ when the spectrum is sharp and misaligned; (ii) Retention: $\lambda_{\text{Low}} \to 1$ when the spectrum is flat or highly aligned; (iii) Spectral Baseline: $\lambda_{\text{Low}} = \rho$ when alignment is zero; and (iv) Alignment Baseline: $\lambda_{\text{Low}} = \cos \alpha$ when spectral mass is negligible.

The minimal bilinear function satisfying the constraints is:

$$\lambda_{\text{Low}}^{(\ell)} = \rho^{(\ell)} + \left(1 - \rho^{(\ell)}\right) \cos \alpha^{(\ell)}, \quad \lambda_{\text{High}}^{(\ell)} = 1 - \lambda_{\text{Low}}^{(\ell)}. \qquad (8)$$

This formulation ensures that re-emphasizing $W_{\text{Low}}$ occurs only when the update is distributed across weak directions or carries significant fractional energy, properties associated with enhanced OOD robustness. We provide a formal derivation and sensitivity analysis in Appendix F.

# 5. Experiments

We next evaluate MonoSoup across both vision and language domains. On CLIP models, we compare against prior merging methods such as Model Soups and Model-Stock, testing whether MonoSoup can achieve competitive or superior robustness using *only a single* fine-tuned checkpoint. We then extend the evaluation to large language models from the Qwen family (Yang et al., 2025), where we assess their effectiveness on mathematical reasoning and multiple-choice benchmarks. Finally, we study its integration with Wise-FT to examine complementarity with interpolation-based robustness methods. Additional experiments on ConvNeXt (Liu et al., 2022), reported in Appendix H, confirm that the method generalizes beyond Transformer-based architectures. We also verify compatibility with parameter-efficient fine-tuning in Appendix L: MonoSoup yields consistent improvements when applied to LoRA-merged checkpoints, with gains that are attenuated but principled — a direct consequence of the rank structure of the low-rank update.

## 5.1. Merging Vision Transformers

We begin with CLIP ViT-B/32 models fine-tuned on ImageNet, the standard testbed for soup-based approaches. In-distribution (ID) accuracy is measured on ImageNet-1K, while out-of-distribution (OOD) robustness is assessed on five natural shifts: ImageNet-V2, ImageNet-R, ImageNet-Sketch, ImageNet-A, and ObjectNet. We use the 70 CLIP ViT-B/32 checkpoints released by Wortsman et al. (2022a). Since presenting results for all 70 models would be impractical, we focus on four representative cases: the checkpoint with the highest ID accuracy ($ID^+$), the lowest ID accuracy ($ID^-$), the highest OOD accuracy ($OOD^+$), and the lowest OOD accuracy ($OOD^-$). This selection allows us to illustrate how MonoSoup behaves across both strong and weak checkpoints. We report results using both a fixed threshold ($R = 0.8$) and the hyperparameter-free MonoSoup. We ablate the fixed threshold in Appendix I and subsection 5.4.

Table 1 presents the results for vision transformers. We also report the number of checkpoints required by each method in the *Cost* column. MonoSoup consistently matches or surpasses multi-model approaches while requiring only a single checkpoint. On the strongest OOD model, Avg. OOD increases from 50.67% to 51.60%, surpassing Greedy Soup without aggregating 70 models. The automated MonoSoup reaches 50.91% without manual tuning. On weaker checkpoints, MonoSoup recovers collapsed representations, improving Avg. OOD by +7.9% for the worst-ID model and +7.5% for the worst-OOD model. Notably, automated MonoSoup follows these gains closely across all representative checkpoints, demonstrating its reliability as a *data-free* and *hyperparameter-free* solution.

*Table 1.* **Merging methods on CLIP ViT-B/32.** Metrics are Top-1 ID and average OOD accuracy for best ($+$) and worst ($-$) models. *Cost* indicates required checkpoints (Soups $\leq 70$, ModelStock 2, MonoSoup 1). MonoSoup matches or exceeds multi-model baselines using a single checkpoint, recovering up to $\sim 8\%$ OOD accuracy on collapsed models. **Top:** Our variants. **Bottom:** Comparison vs. ModelStock.

**General Baselines**

|  | ID | OOD | Cost |
|---|---|---|---|
| Initialization | 75.4% | 46.2% | – |
| Uniform Model Soup | 79.9% | 51.4% | 70 |
| Greedy Model Soup | **81.0%** | 50.7% | 70 |
| FT model ($OOD^+$) | 78.11% | 50.67% | – |
| FT model ($OOD^-$) | 76.53% | 36.71% | – |
| FT model ($ID^+$) | 80.38% | 47.96% | – |
| FT model ($ID^-$) | 74.99% | 38.64% | – |

**Our Single-Model Methods**

| Model | MonoSoup ($R = 0.8$) | | MonoSoup | |
|---|---|---|---|---|
|  | ID | OOD | ID | OOD |
| $OOD^+$ | 78.29% | **51.60%** | 78.21% | 50.91% |
| $OOD^-$ | 78.55% | 44.21% | 78.03% | 42.78% |
| $ID^+$ | 80.03% | 49.95% | 80.34% | 48.94% |
| $ID^-$ | 77.76% | 46.54% | 77.38% | 45.22% |

**Pairwise Methods**

| Model Pair | ModelStock | | MonoSoup | |
|---|---|---|---|---|
|  | ID | OOD | ID | OOD |
| $ID^+, OOD^+$ | 79.39% | 50.53% | 80.10% | 51.37% |
| $ID^+, OOD^-$ | 78.43% | 49.39% | 78.87% | 48.37% |
| $ID^+, ID^-$ | 78.32% | 50.63% | 79.02% | 50.12% |
| $OOD^+, OOD^-$ | 76.76% | 48.41% | 78.44% | 49.26% |
| $OOD^+, ID^-$ | 77.49% | 51.02% | 78.05% | 50.48% |
| $ID^-, OOD^-$ | 78.09% | 47.81% | 78.94% | 47.79% |

To assess robustness beyond the four representative checkpoints, Figure 4 reports the effect of MonoSoup across all 70 fine-tuned models. The dominant upward trend confirms that OOD gains are systematic across the full range of hyperparameter configurations, with at most marginal ID tradeoffs in a small number of cases.

When applied to pairs of fine-tuned models, MonoSoup achieves a better balance between ID and OOD performance compared to ModelStock. This is especially pronounced in the pair of ($ID^+$, $OOD^+$) where MonoSoup dominates on both objectives. This shows that the benefits of the proposed method extend beyond the single-checkpoint setting.

## 5.2. Merging Large Language Models

To test the generality of MonoSoup beyond vision, we evaluate it on large language models. Specifically, we fine-tune multiple variants of the Qwen3-0.6B model, each with different hyperparameter configurations, such as learning rates,

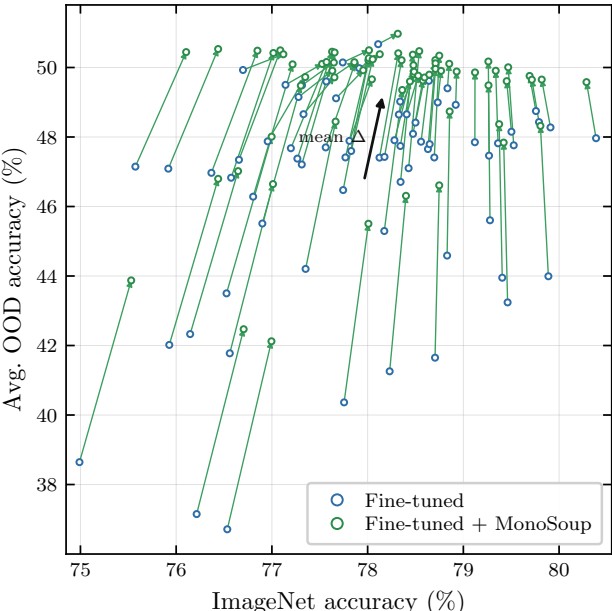

*Figure 4.* **Application of MonoSoup on 70 CLIP checkpoints.** MonoSoup improves OOD accuracy across all 70 CLIP ViT-B/32 checkpoints by Wortsman et al. (2022a) while maintaining competitive ID performance. Each arrow originates at the fine-tuned checkpoint's performance and points to the post-MonoSoup result; the dominant upward trend confirms that OOD gains are systematic across all hyperparameter configurations, with at most marginal ID tradeoffs in a small number of cases.

*Table 2.* **Performance of Qwen3-0.6B on mathematical benchmarks.** We compare our single-model methods (**MonoSoup** and **MonoSoup** with fixed variance threshold ($R = 0.8$)) against the LiNeS baseline and pairwise merging (ModelStock) across three training settings (M-1, M-2, M-3; details in Appendix E). **MonoSoup** and **MonoSoup** ($R = 0.8$) consistently outperform the fine-tuned baseline and LiNeS, showing the largest gains on the challenging GSM$_{Plus}$ and GSM$_{Plat}$ tasks. Notably, it matches or surpasses ModelStock while using *only a single* checkpoint.

| Setting | Configuration Method | GSM8K | GSM$_{Plus}$ | GSM$_{Plat}$ | SciQ | MMLU-P |
|---|---|---|---|---|---|---|
| *Reference* | QWEN3-0.6B-BASE | 52.6 | 22.5 | 50.1 | 92.6 | 33.7 |
| **M-1** *(Linear)* | Standard | 55.8 | 29.5 | 55.6 | 94.6 | 35.6 |
| | + LiNeS | 56.2 | 30.1 | 56.3 | 94.9 | 36.7 |
| | + MonoSoup $R = 0.8$ **(Ours)** | **56.7** | **30.3** | **56.7** | 95.1 | 37.2 |
| | + MonoSoup **(Ours)** | **56.9** | 30.2 | 56.8 | 95.3 | 37.5 |
| **M-2** *(Cosine)* | Standard | 56.1 | 30.8 | 58.5 | 94.5 | 35.9 |
| | + LiNeS | 56.5 | 31.4 | 58.9 | 95.2 | 36.1 |
| | + MonoSoup $R = 0.8$ **(Ours)** | 56.6 | **31.7** | 59.3 | 95.1 | **36.6** |
| | + MonoSoup **(Ours)** | **56.8** | 31.9 | 59.4 | 95.3 | 36.9 |
| **M-3** *(Ext.)* | Standard | 55.9 | 30.6 | 57.3 | 93.5 | 34.8 |
| | + LiNeS | 56.3 | 30.8 | 58.1 | 93.8 | 35.2 |
| | + MonoSoup $R = 0.8$ **(Ours)** | 56.6 | 31.4 | 58.8 | 94.2 | 35.5 |
| | + MonoSoup **(Ours)** | **56.9** | 31.7 | 59.2 | **94.5** | 35.8 |
| **Pairwise** | ModelStock (M-1, M-2) | 56.6 | 31.5 | 59.0 | 94.9 | 36.8 |
| | ModelStock (M-1, M-3) | 56.4 | 31.3 | 58.7 | 94.8 | 36.4 |
| | ModelStock (M-2, M-3) | 56.5 | 31.6 | 58.9 | 95.2 | 36.7 |

training epochs, and schedules. Further details are provided in Appendix E. All variants are trained on a mixture spanning mathematical reasoning and multiple-choice question answering: MetaMathQA (Yu et al., 2024), which augments GSM8K (Cobbe et al., 2021) and MATH (Hendrycks et al., 2021c); DeepMind-AquaRat (Ling et al., 2017); and the multiple-choice datasets OpenBookQA (Mihaylov et al., 2018) and SciQ (Welbl et al., 2017).

For evaluation, we propose a benchmark that spans a wide spectrum of reasoning difficulty. It includes GSM8K and SciQ, which overlap with the training mixture and are treated as in-distribution tasks, as well as GSM$_{Plus}$ (Li et al., 2024), GSM8K$_{Platinum}$ (Vendrow et al., 2025), and MMLU-Pro-Math (Wang et al., 2024b), which probe advanced or adversarial reasoning skills not explicitly covered during training and thus serve as out-of-distribution evaluations. While this ID/OOD split is less rigid than in vision benchmarks such as CLIP, the increasing task difficulty provides an analogous way to assess robustness and generalization in the language domain.

The results are presented in Table 2 and demonstrate consistent improvements across all fine-tuned variants. MonoSoup improves over the baseline Qwen3-0.6B models and surpasses LiNeS across every benchmark, with the

largest gains observed on GSM$_{Plus}$ and GSM8K$_{Platinum}$ (+9.2 points each). Compared to ModelStock, which remains competitive when merging pairs of models, MonoSoup matches or exceeds its performance while requiring only a single checkpoint. These findings mirror the results on CLIP: MonoSoup enhances robustness and generalization *without reliance on ensembles*, scaling naturally when multiple models are available but remaining highly effective in the single-checkpoint setting.

We further evaluate MonoSoup along two complementary axes of distributional shift in Appendix K: a cross-lingual shift on MGSM (Shi et al., 2023), where the model is evaluated on non-English languages unseen during fine-tuning, and a cross-domain shift on ARC, HellaSwag, and MMLU, where the target capabilities are entirely absent from the mathematics-focused training data. Across both axes, spectral reweighting not only recovers but refines the multilingual and general-purpose representations eroded by fine-tuning, surpassing even the pretrained baseline on average across languages.

### 5.3. Integration with Wise-FT

We next study how MonoSoup interacts with Wise-FT (Wortsman et al., 2022b), which linearly interpolates between pre-trained and fine-tuned weights to produce a continuum of models tracing the ID–OOD trade-off. This setting allows us to test whether MonoSoup can serve as a stronger endpoint for interpolation-based robustness methods. We compare against two baselines: Wise-FT alone, and LiNeS (Wang et al., 2025a), a post-training technique that

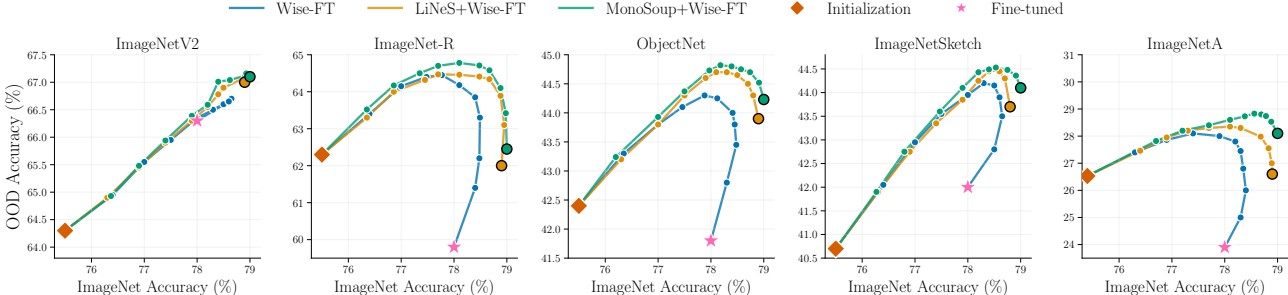

*Figure 5.* MonoSoup integrated with Wise-FT on CLIP ViT-B/32. MonoSoup improves ID and OOD accuracy across individual checkpoints. When combined with Wise-FT, the Pareto fronts consistently dominate those of Wise-FT and LiNeS, showing that MonoSoup provides a stronger endpoint for interpolation-based robustness.

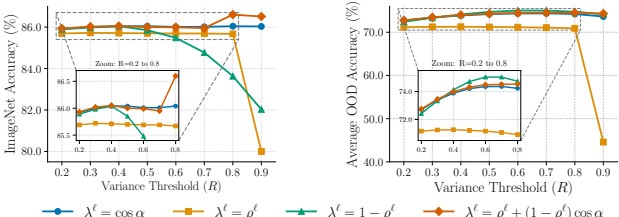

*Figure 6.* **Component Analysis.** Effect of varying the variance threshold $R$ and the contributions of each term in the coefficient $\lambda^\ell = \lambda^\ell_{\text{low}}$ on CLIP ViT-L/14. Results are stable across a wide range of $R$ values, and both the spectral decay and cosine overlap components contribute meaningfully to the final balance between ID and OOD performance.

linearly scales fine-tuning updates according to layer depth. Unlike MonoSoup, which is fully data-free and adapts coefficients at the level of individual subspaces within each layer to account for heterogeneous layer dynamics, LiNeS employs a single coefficient per transformer block, neglecting the distinct fine-tuning dynamics of attention versus feedforward layers (Yang et al., 2024), and requires labeled data for hyperparameter selection.

We report results averaged over all 70 checkpoints in Figure 5. Consistent with the results of Figure 4, MonoSoup applied in isolation improves both ID and OOD performance over the fine-tuned baseline across all five distribution shifts, confirming that spectral reweighting alone yields a strictly better checkpoint. When used as the starting point for Wise-FT interpolation, the resulting Pareto fronts consistently dominate those of Wise-FT alone and LiNeS across all evaluated datasets. This demonstrates that MonoSoup is complementary to interpolation-based methods: by providing a stronger, data-free base checkpoint, it further amplifies the benefits of existing robustness techniques.

### 5.4. Analysis and Discussion

We analyze two design aspects of MonoSoup: the effect of the spectral energy threshold $R$ and the role of each component in the mixing rule. We focus on the fixed-$R$ formulation from Equation 2 to shed light on the role of the spectral par-

tition and the sensitivity of MonoSoup to this design choice. Varying $R$ controls how much of the fine-tuning update is assigned to the high-energy subspace. The results on CLIP ViT-L/14, shown in the left panel of Figure 6, reveal three clear patterns. When $R$ is too small, too much spectral mass is discarded, which hurts both ID and OOD performance. Very large values of $R$ keep almost all directions, saturating improvements and sometimes leading to collapse. Intermediate values around $0.7$–$0.85$ achieve the best balance, confirming that low-energy directions are important but must be modulated relative to dominant task-specific updates.

We also ablate the two signals in our mixing rule: spectral decay $\rho^\ell$ and alignment $\cos\alpha^\ell$. The right panel of Figure 6 shows that relying only on $\rho^\ell$ preserves ID accuracy but brings little OOD improvement, while relying only on $\cos\alpha^\ell$ improves OOD on weaker checkpoints but can reduce ID. Uniform mixing gives inconsistent results, and keeping only the high- or low-energy components degrades one side of the trade-off. In contrast, combining $\rho^\ell$ and $\cos\alpha^\ell$ consistently yields the strongest OOD performance without sacrificing ID, and remains stable across a wide range of $R$. These results validate the design of MonoSoup: both spectral decay and alignment provide complementary signals, and together they enable a principled way to retain the benefits of low-energy directions without undermining task-specific adaptations. We further verify that MonoSoup applies genuinely anisotropic scaling across layers, rather than collapsing to scalar interpolation; see Figure J.

## 6. Related Work

**Representation Collapse and Robust Fine-Tuning.** The prevalent pre-train-then-finetune paradigm often leads to a degradation of a model's general-purpose knowledge, resulting in a decline in out-of-distribution (OOD) performance (Kumar et al., 2022). This phenomenon, termed *representation collapse* (Aghajanyan et al., 2021), has motivated a significant body of research focused on making the fine-tuning process more robust. Such methods typically regularize the fine-tuning process to preserve the valuable features learned

during pre-training, thereby improving OOD generalization (Gouk et al., 2021; Zhang et al., 2022; Razdaibiedina et al., 2023; Lee et al., 2023; Goyal et al., 2023; Wortsman et al., 2022b; Mao et al., 2024; Nam et al., 2024; Oh et al., 2024). While effective, these approaches intervene directly in the computationally expensive fine-tuning stage, motivating the exploration of more efficient, post-hoc alternatives.

**Mode Connectivity and Post-hoc Merging.** An alternative line of work focuses on post-hoc manipulation of model weights, a practice theoretically grounded in the properties of the neural network loss landscape. Seminal works showed that distinct solutions found by separate training runs can be connected by a non-linear path of low loss (Garipov et al., 2018; Draxler et al., 2018). More critically for fine-tuning, Frankle et al. (2020) demonstrated the existence of *linear mode connectivity* between models that share the same pre-trained initialization. This property enables simple yet powerful techniques like weight averaging. By interpolating the parameters of multiple fine-tuned checkpoints, these methods have been shown to find wider, more robust optima (Izmailov et al., 2018; Wortsman et al., 2021), leading to improved in-distribution (Wortsman et al., 2022a; Jang et al., 2024) and out-of-distribution performance (Wortsman et al., 2022b; Ramé et al., 2022; 2023) without requiring additional inference costs.

**Unifying Principles of Successful Merging.** Recent analyses of these merging techniques have revealed a unifying principle: successful weight-space ensembling reinforces dominant directions in the weight space that encode shared, task-relevant signals, while simultaneously suppressing noisy or misaligned directions that harm generalization (Wortsman et al., 2022a; Jang et al., 2024; Ramé et al., 2023). This insight has inspired the development of more sophisticated merging strategies that explicitly identify and manipulate these core components of the fine-tuning update (Gargiulo et al., 2025; Tang et al., 2025; Wang et al., 2025a). Beyond improving single-model robustness, these principles of weight-space arithmetic have been successfully extended to a broader range of applications, including multi-task learning (Ilharco et al., 2022; 2023; Dimitriadis et al., 2023; 2025; Yadav et al., 2023; Giraldo et al., 2025) and multi-objective alignment (Ramé et al., 2024; Zhong et al., 2024).

## 7. Conclusion

In this paper, we introduced MonoSoup, a data-free method that reweights the spectral components of fine-tuning updates to improve both in-distribution accuracy and out-of-distribution robustness from a single checkpoint. Unlike prior approaches that depend on ensembles of fine-tuned models or carefully aligned pairs, MonoSoup compresses their benefits into a lightweight, single-model procedure that restores OOD performance even for weak checkpoints.

Experiments on CLIP and Qwen benchmarks show its effectiveness across vision and language domains, demonstrating that robust gains are possible without the computational and storage overhead of multi-model methods. Looking forward, extending our approach beyond vision and language to other modalities offers a promising direction. Overall, our results highlight that the benefits of model soups can be retained, even strengthened, without the burden of maintaining large ensembles, making MonoSoup a practical plug-and-play tool for reliable model deployment.

## Impact Statement

This paper presents MonoSoup, a post-hoc, data-free editing procedure that improves the in-distribution / out-of-distribution trade-off of fine-tuned models from a *single* checkpoint. Its principal societal contribution is a substantial reduction in the compute and storage cost of robust deployment: as shown in Table 7, MonoSoup matches or exceeds multi-checkpoint baselines such as ModelSoups while reducing wall-clock cost by $\sim 10^3$–$10^4 \times$ and storage by an order of magnitude, displacing on the order of $240$ GPU-hours per deployment for a $14$B-parameter language model. Because public model repositories typically expose only a single best-performing checkpoint per task, operating on one checkpoint also lowers the barrier to robust fine-tuning for practitioners without the budget to reproduce large hyperparameter sweeps, and yields a deterministic, auditable edit that we view as a positive externality for the broader robustness literature. We emphasise, however, that the robustness improvements demonstrated here concern *natural* distribution shifts and should not be interpreted as guarantees against adversarial perturbations underlying our spectral analysis; MonoSoup is therefore not a substitute for application-specific safety evaluation.

## Acknowledgments

We thank Alessandro Favero, Ke Wang, Amel Abdelraheem, and Seyed-Mohsen Moosavi-Dezfooli for insightful discussions and valuable feedback that helped shape this work.

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

*Table 3.* A comprehensive comparison of single-model merging methods on CLIP ViT-B/32. Performance metrics are presented for ImageNet and the average of five OOD datasets across fine-tuned models utilizing *LP initialization* released by Wortsman et al. (2022a). For Wise-FT, we sweep the interpolation coefficient $\alpha$ and report the best-performing setting with respect to Avg. OOD.

| Method | Linear Probe initialization | |
|---|---|---|
| | ID (ImageNet) | Avg. OOD |
| **Baseline** | | |
| CLIP LP Initialization | 75.40% | 46.20% |
| **Fine-Tuned Models** | | |
| FT model (Best Avg. OOD) | 78.11% | 50.67% |
| FT model (Worst Avg. OOD) | 76.53% | 36.71% |
| FT model (Best ID) | 80.38% | 47.96% |
| FT model (Worst ID) | 74.99% | 38.64% |
| **LiNeS** | | |
| LiNeS+FT model (Best Avg. OOD) | 78.25% | 51.56% |
| LiNeS+FT model (Worst Avg. OOD) | 78.13% | 41.33% |
| LiNeS+FT model (Best ID) | 80.46% | 49.10% |
| LiNeS+FT model (Worst ID) | 77.31% | 44.96% |
| **Wise-FT** | | |
| Wise-FT+FT model (Best Avg. OOD) | 78.12% | 51.54% |
| Wise-FT+FT model (Worst Avg. OOD) | 77.11% | 45.2% |
| Wise-FT+FT model (Best ID) | 78.73% | 49.12% |
| Wise-FT+FT model (Worst ID) | 78.00% | 45.75% |
| **Our Proposed Method** | | |
| MonoSoup +FT model (Best Avg. OOD) | 78.29% | 51.60% |
| MonoSoup +FT model (Worst Avg. OOD) | 78.55% | 44.21% |
| MonoSoup +FT model (Best ID) | 80.03% | 49.95% |
| MonoSoup +FT model (Worst ID) | 77.76% | 46.54% |

## A. Comparison with single-model merging Methods

**Linear Probing initialization (LP init).** Table 3 presents a comprehensive analysis of single-model merging methods for models with *Linear Probing initialization (LP init)*. MonoSoup significantly improves the M-14 model, which has the worst average OOD performance, increasing OOD accuracy from $36.71\%$ to $44.21\%$ (a gain of $7.5\%$) and ID accuracy by $2.02\%$. Furthermore, it enhances the M-31 model, which has the worst ID performance, achieving a $2.77\%$ accuracy gain.

## B. Zero-shot initialization (ZS init)

Table 4 presents a comprehensive analysis of single-model merging methods for models with *zero-shot initialization (ZS init)*. We use two publicly available *ZS*-initialized checkpoints from Jang et al. (2024). Our proposed MonoSoup achieves improvements of $6.4\%$ in average OOD accuracy and $0.8\%$ and $0.6\%$ in ID accuracy across the respective experimental configurations. For comparison with *soup*-model merging methods in the two-checkpoint scenario, we apply our method to the average of two checkpoints, resulting in performance gains of $2.9\%$ over ModelStock, $3.0\%$ over GreedySoup, and $1.8\%$ over UniformSoups.

## C. Connection between $R$ and $\cos\alpha$

Let $r = \text{rank}(\boldsymbol{W})$ and let $\sigma_j = \sigma_j(\boldsymbol{W})$ denote the $j$-th singular value of $\boldsymbol{W}$ for $j \in \{1, \dots, r\}$. The Frobenius norm of $\boldsymbol{W}$ can be expressed in terms of its singular values as $\|\boldsymbol{W}\|_F^2 = \sum_{j=1}^r \sigma_j^2$. Given a target variance capture ratio $R \in [0, 1]$, we define the truncation index $k$ as

$$k = \underset{j \in \{1, \dots, r\}}{\arg\min} \left\{ j \,\middle|\, \frac{\sum_{s=1}^j \sigma_s^2}{\|\boldsymbol{W}\|_F^2} \geq R \right\}. \tag{9}$$

Let $P_k = \sum_{s=1}^k \sigma_s^2 / \|\boldsymbol{W}\|_F^2$ denote the actual fraction of variance captured by the truncated matrix $\boldsymbol{W}_{\text{High-Energy}}$. By the

*Table 4.* Comparison of single-model merging methods on CLIP ViT-B/32 with *zero-shot initialization (ZS init)*. We use the two publicly available checkpoints from Jang et al. (2024) and use their reported numbers for the Model Soups baselines with 48 models.

| Method | | Zero-Shot Initialization | | |
|---|---|---|---|---|
| | | ID (ImageNet) | Avg. OOD | Cost |
| *Baselines* | | | | |
| Initialization | | 63.3% | 48.5% | 0 |
| Vanilla FT 1 | | 78.1% | 46.7% | 1 |
| Vanilla FT 2 | | 78.3% | 46.9% | 1 |
| *LiNeS* | | | | |
| LiNeS ($\alpha$=0.1, $\beta$=0.9) + FT 1 | | 78.7% | 52.2% | 1 |
| LiNeS ($\alpha$=0.5, $\beta$=0.5) + FT 1 | | 78.9% | 51.1% | 1 |
| LiNeS ($\alpha$=0.1, $\beta$=0.9) + FT 2 | | 78.5% | 51.9% | 1 |
| LiNeS ($\alpha$=0.5, $\beta$=0.5) + FT 2 | | 79.0% | 51.1% | 1 |
| *Wise-FT* | | | | |
| Wise-FT+ FT 1 | | 78.8% | 52.5% | 1 |
| Wise-FT+ FT 2 | | 78.8% | 52.6% | 1 |
| *Prior Soups-Merging Methods* | | | | |
| Uniform Model Soup | | 79.7% | 52.0% | 48 |
| Greedy Model Soup | | 80.4% | 50.8% | 48 |
| ModelStock | | 79.8% | 50.9% | 2 |
| *Our Proposed Method* | | | | |
| MonoSoup w/ Vanilla FT 1 | | 78.9% | 53.1% | 1 |
| MonoSoup w/ Vanilla FT 2 | | 78.9% | 53.2% | 1 |
| MonoSoup w/ Avg. of FT 1&2 | | 79.0% | 53.8% | 2 |

definition of $k$, we have

$$P_k \geq R.$$

**Lemma 1.** *If $k > 1$, then*

$$P_k - \frac{\sigma_k^2}{\|\boldsymbol{W}\|_F^2} < R \leq P_k.$$

*Proof.* Since $k$ is the minimum index satisfying the variance threshold, we have that $k-1$ does not satisfy it, i.e.,

$$\frac{\sum_{s=1}^{k-1} \sigma_s^2}{\|\boldsymbol{W}\|_F^2} < R.$$

Observing that $\sum_{s=1}^{k-1} \sigma_s^2 = \sum_{s=1}^{k} \sigma_s^2 - \sigma_k^2$, we obtain

$$P_k - \frac{\sigma_k^2}{\|\boldsymbol{W}\|_F^2} < R.$$

Combined with $P_k \geq R$, the result follows. $\square$

Now, we establish the relationship with $\cos\alpha$. By the orthogonal decomposition $\boldsymbol{W} = \boldsymbol{W}_{\text{High-Energy}} + \boldsymbol{W}_{\text{Low-Energy}}$ and the Pythagorean theorem in Frobenius norm, we have

$$\|\boldsymbol{W}\|_F^2 = \|\boldsymbol{W}_{\text{High-Energy}}\|_F^2 + \|\boldsymbol{W}_{\text{Low-Energy}}\|_F^2.$$

Since $\cos\alpha = \|\boldsymbol{W}_{\text{Low-Energy}}\|_F / \|\boldsymbol{W}\|_F$, we obtain

$$\cos^2\alpha = \frac{\|\boldsymbol{W}_{\text{Low-Energy}}\|_F^2}{\|\boldsymbol{W}\|_F^2} = \frac{\|\boldsymbol{W}\|_F^2 - \|\boldsymbol{W}_{\text{High-Energy}}\|_F^2}{\|\boldsymbol{W}\|_F^2} = 1 - P_k.$$

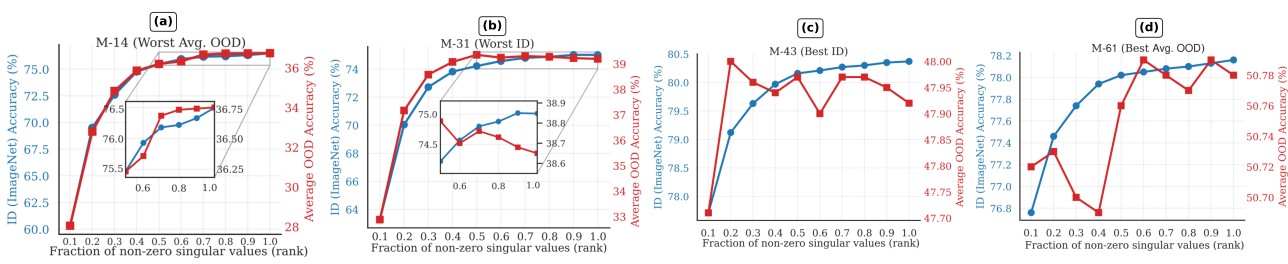

*Figure 7.* The task vector rank consistently enhances performance on **both** ID and OOD benchmarks. The x-axis represents the rank of the task vector, with blue curves indicating ID accuracy and red curves depicting average OOD accuracy.

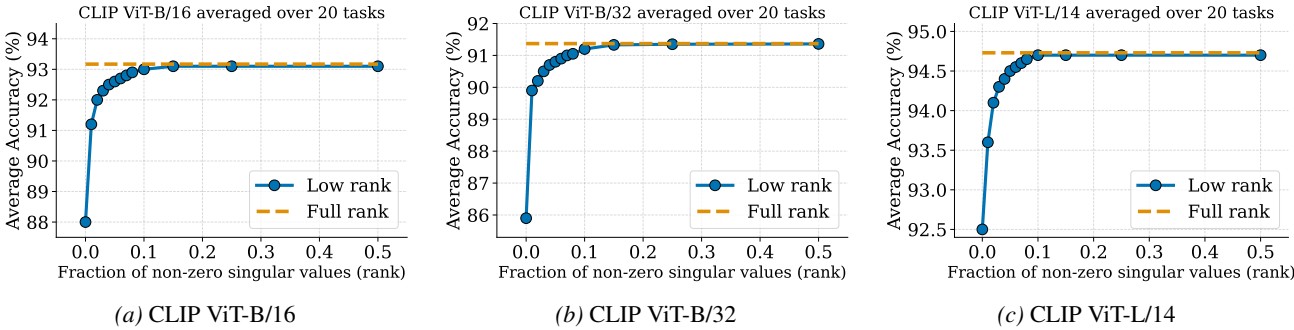

*(a)* CLIP ViT-B/16        *(b)* CLIP ViT-B/32        *(c)* CLIP ViT-L/14

*Figure 8.* Mean absolute accuracy of the CLIP ViT-{B/32, B/16, L/14} models across increasing fractions of retained singular components, averaged over 20 tasks released by (Wang et al., 2024a). The yellow line represents the average accuracy of the original fine-tuned models with full-rank task matrices, while the blue line shows the accuracies using low-rank approximations.

**Theorem 1.** *The angle parameter $\alpha$ satisfies the following bounds:*

$$\max\left(0, 1 - R - \frac{\sigma_k^2}{\|\boldsymbol{W}\|_F^2}\right) < \cos^2 \alpha \leq 1 - R.$$

*Proof.* From the relationship $\cos^2 \alpha = 1 - P_k$ and the inequality $R \leq P_k$, we immediately obtain

$$\cos^2 \alpha \leq 1 - R.$$

For the lower bound, when $k > 1$, we have $P_k - \sigma_k^2/\|\boldsymbol{W}\|_F^2 < R$, which yields

$$1 - \cos^2 \alpha - \frac{\sigma_k^2}{\|\boldsymbol{W}\|_F^2} < R,$$

and therefore

$$\cos^2 \alpha > 1 - R - \frac{\sigma_k^2}{\|\boldsymbol{W}\|_F^2}.$$

Since $\cos^2 \alpha \geq 0$, the lower bound becomes

$$\cos^2 \alpha > \max\left(0, 1 - R - \frac{\sigma_k^2}{\|\boldsymbol{W}\|_F^2}\right).$$

$\square$

# D. Low-Energy Directions

In this section, we present additional observations that build upon and extend the analyses discussed in Figure 3.

In Figure 7, we demonstrate that increasing task vector rank consistently enhances performance on both ID and OOD benchmarks. Our analysis reveals a clear positive correlation between retaining small singular values, *i.e.*, Low-Energy, and improved generalization performance across diverse model configurations, spanning from high-performing ID models to those with poor average OOD performance. Notably, even when preserving 95% of singular values, performance degradation occurs on both ID and OOD tasks, demonstrating that truncation-based approaches fail to enhance generalization and, counterintuitively, that low-energy components contain critical information for robust performance.

This finding contrasts sharply with our observations on smaller-scale downstream tasks. When replicating truncation experiments across *downstream task arithmetic* operations involving the 20 tasks proposed by (Wang et al., 2024a), the task matrices exhibit pronounced low-rank properties, corroborating previous findings that a limited subset of task vectors can accurately represent each layer's functionality (see Figure 8). Remarkably, retaining only 5% of singular components for each task yields mean accuracy comparable to the original fine-tuned models. This suggests that 95% of singular components in each layer matrix can be removed without significant performance degradation on these smaller-scale benchmarks.

## E. Qwen fine-tuning

In this section, we detail the fine-tuning procedure for Qwen. We perform *full-parameter fine-tuning* using the AdamW optimizer (Loshchilov & Hutter, 2019) with a batch size of 32. To mitigate train–test context mismatch, we fix the context length at 2,048 tokens. We evaluate three specific model variants: M-1, which utilizes a linear learning rate schedule; M-2, which employs a cosine schedule; and M-3, which uses a cosine schedule but is trained for two additional epochs compared to the others.

## F. Derivation and Interpretability of Coefficients

The coefficient $\lambda_{\text{Low}} = f(\rho, \cos \alpha)$ is derived from a set of logical boundary conditions rather than heuristic selection. We seek a smooth function $f : [0, 1]^2 \to [0, 1]$ that behaves predictably at the limits of the spectral and geometric signals:

- $f(0, 0) = 0$: Suppress low-energy components for sharp, misaligned spectra.

- $f(1, c) = 1$ and $f(\rho, 1) = 1$: Retain components if either signal is maximal.

- $f(\rho, 0) = \rho$: Fall back to the spectral decay ratio when alignment is absent.

- $f(0, \cos \alpha) = \cos \alpha$: Rely on alignment when the spectral mass $\rho$ is negligible.

Assuming the interaction between $\rho$ and $\cos \alpha$ is bilinear—representing the simplest smooth interaction—these four constraints *uniquely determine* the mapping:

$$f(\rho, \cos \alpha) = \rho + \cos \alpha - \rho \cos \alpha = \rho + (1 - \rho) \cos \alpha. \tag{10}$$

**Sensitivity and Stability.** MonoSoup is a one-shot update that is 1-Lipschitz in both arguments:

$$\frac{\partial f}{\partial \rho} = 1 - \cos \alpha \in [0, 1], \quad \frac{\partial f}{\partial \cos \alpha} = 1 - \rho \in [0, 1]. \tag{11}$$

## G. Centered Kernel Alignment analysis

To investigate the representational dynamics induced by MonoSoup, we analyze the internal feature spaces of the model components using Centered Kernel Alignment (CKA) (Kornblith et al., 2019). While our methodology operates directly on model weights, this analysis serves to verify that spectral reweighting translates to meaningful preservation of pre-trained features on out-of-distribution (OOD) data without collapsing the specialized signals acquired during fine-tuning. By measuring the similarity between the hidden representations of edited checkpoints and the original pre-trained backbone, we provide empirical grounding for the roles of high- and low-energy subspaces in the generalization-specialization trade-off.

For each transformer block $\ell$, we compare the hidden features of: (a) Pre-trained, (b) fine-tuned, (c) MonoSoup, (d) High-only ($\boldsymbol{W}_{\text{High}}$), (e) Low-only ($\boldsymbol{W}_{\text{Low}}$). We compute linear CKA (Kornblith et al., 2019) to the pre-trained features on an unlabeled set from ImageNet-1K and OOD set.

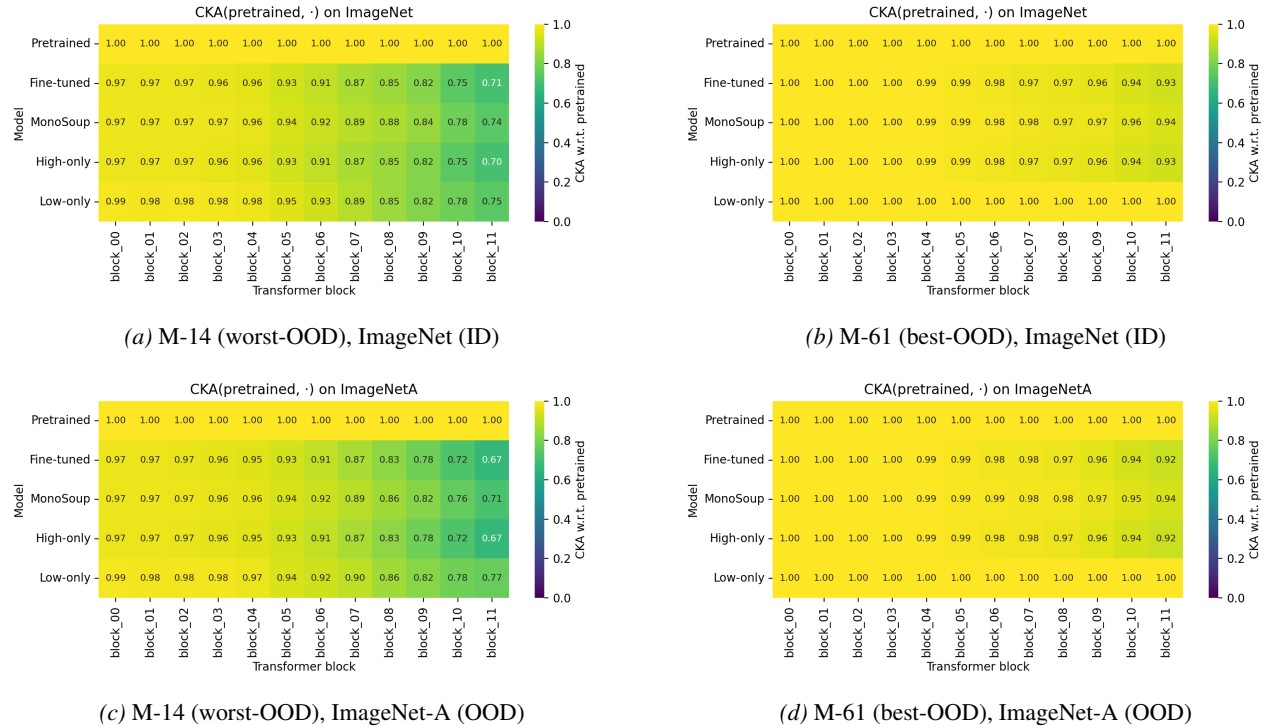

*(a)* M-14 (worst-OOD), ImageNet (ID)

*(b)* M-61 (best-OOD), ImageNet (ID)

*(c)* M-14 (worst-OOD), ImageNet-A (OOD)

*(d)* M-61 (best-OOD), ImageNet-A (OOD)

*Figure 9.* Feature-space alignment maps (CKA) between edited checkpoints and the pre-trained backbone, arranged by *model* (columns: M-14 worst-OOD, M-61 best-OOD) and *evaluation distribution* (rows: ImageNet ID, ImageNet-A OOD). On the weak checkpoint M-14, fine-tuning distorts deep layers most severely under OOD (panel c); MonoSoup partially restores the pre-trained representation in those layers, mirroring the small disruption already present on the strong checkpoint M-61.

This analysis examines whether MonoSoup preserves pre-trained representations on OOD data while retaining task-specific specialization on ID data. Concretely, we expect the edited checkpoint's features to remain closer to the pre-trained backbone than the fully fine-tuned model on OOD inputs, without reverting to the pre-trained solution on ID. To this end, we plot the layer-by-layer CKA similarity to the pre-trained features for five model variants — (a) Pre-trained, (b) Fine-tuned, (c) MonoSoup, (d) High-only ($\mathbf{W}_{\text{High}}$), (e) Low-only ($\mathbf{W}_{\text{Low}}$) — on both ID and OOD data, for the worst- and best-OOD checkpoints in Figure 9.

Due to the computational cost of calculating the CKA for each transformer block, we conduct this experiment using 25 batches of size 256 on ImageNet (ID) data and the most challenging OOD dataset among ImageNet distribution shifts, which is ImageNet-A (OOD). This setup ensures that all samples from ImageNet-A are included within the specified number of batches and samples per batch.

As a validation baseline, we first plot the pretrained row, confirming a constant CKA value of $1.00$. For the 'Fine-tuned' model (M-14), early blocks (0–3) maintain high similarity ($\approx 0.97$), while middle blocks (4–6) decline slightly from $0.96$ to $0.93$. However, the deeper blocks (7–11) deviate progressively, falling from $0.87$ to $0.67$. This contrasts sharply with the strong model (best-OOD, right panel), which retains significantly more pretrained structure in its deepest layers; notably, Block 11 maintains a CKA of $0.92$ in the strong model compared to $0.67$ in the fine-tuned version. Finally, MonoSoup bridges this gap: while its early blocks mirror the fine-tuned model, its deeper blocks consistently exhibit higher CKA values (e.g., Block 11 reaches $0.71$ vs. $0.67$), indicating better preservation of pretrained features.

MonoSoup preserves the integrity of shallow and mid layers while noticeably realigning the deepest layers toward the pretrained ImageNetA representation. This visually confirms that MonoSoup partially de-specializes the model, enhancing OOD robustness while retaining the essential fine-tuned signal. In contrast, the *High-only row closely mirrors the Fine-tuned* model, exhibiting identical values in early blocks ($\approx 0.97$) and a similar decline in deeper blocks (ending near $0.67$). This validates the core insight of our decomposition: high-energy task directions ($\Delta\mathbf{W}_{\text{high}}$) dominate the update, such that $\mathbf{W}_0 + \Delta\mathbf{W}_{\text{high}} \approx \mathbf{W}_1$. These directions capture the primary representational drift from pretraining—drift that empirically correlates with OOD degradation in the weak model (M-14: worst-OOD). Ultimately, this illustrates the *truncate hurts OOD* phe-

nomenon in feature space: retaining only $\Delta \boldsymbol{W}_{\text{high}}$ reproduces the specific representational shifts that compromise robustness.

The Low-only model remains closest to the pretrained representation, showing better OOD performance but poorer in-distribution (ID) results. In contrast, the High-only model aligns with the Fine-tuned model and inherits its OOD vulnerabilities. MonoSoup smoothly interpolates between these extremes, balancing the trade-off. This aligns with our hypothesis: Low-only retains old knowledge at the cost of ID specialization, exhibiting the highest CKA to pretrained but worst ID performance; High-only represents pure task directions, diverging most from pretrained in deep layers and suffering worst OOD; MonoSoup combines these, achieving closer alignment to pretrained than Fine-tuned or High-only on OOD without reverting fully to pretrained features.

On the worst-OOD model like M-14, fine-tuning significantly distorts deep features away from the pretrained representation on OOD data (ImageNetA), with high-energy task directions embodying this distortion. MonoSoup mitigates this by reweighting high- and low-energy updates, notably pulling the deepest layers closer to the pretrained representation (CKA increase of $0.04$–$0.05$), which corresponds to the observed OOD performance gains. Conversely, on stronger backbones where fine-tuning already maintains a high similarity to pretrained features (CKA $\geq 0.9$), MonoSoup's adjustments and accuracy improvements are naturally smaller. This exemplifies the principle that MonoSoup is most effective when fine-tuning begins to disrupt the pretrained representation, as demonstrated by two models and their corresponding CKA heatmaps.

Recall that $\boldsymbol{W}_{\text{Low}}^{(\ell)}$ is the component whose magnitude is modulated by $\cos \alpha^{(\ell)}$ through $\lambda_{\text{low}}^{(\ell)}$ in Equation 8. The fact that the Low-only model remains closest to the pre-trained representation (highest CKA) and improves OOD at the cost of ID, while High-only mirrors the fine-tuned representation and inherits its OOD vulnerabilities, empirically validates our interpretation of $\cos \alpha^{(\ell)}$ as a *pre-training preservation signal*: increasing $\lambda_{\text{low}}^{(\ell)}$ (hence $\cos \alpha^{(\ell)}$ ) shifts the representation toward the pre-trained solution on OOD inputs in exactly the way predicted by our CKA analysis.

# H. ConvNeXt

To verify that MonoSoup generalises beyond Transformer-based architectures, we evaluate on ConvNeXt-Small (Liu et al., 2022) pretrained on ImageNet-22k and fine-tuned on ImageNet-1k. Table 5 reports both the automated (entropy-based effective rank) and fixed-threshold ($R=0.8$) variants. Both improve over the fine-tuned baseline on ID and OOD accuracy, with the fixed-threshold variant achieving the largest gains ($+0.40$ pp ID, $+0.85$ pp OOD).

*Table 5.* Performance of MonoSoup on ConvNeXt-Small (IN-22k $\rightarrow$ IN-1k).

| Method | ID (ImageNet) | Avg. OOD |
|---|---|---|
| Pretrained (IN-22k) | 82.17% | 52.88% |
| Fine-tuned (IN-1k) | 85.17% | 55.85% |
| MonoSoup (auto) | 85.19% | 56.07% |
| MonoSoup ($R=0.8$) | **85.57%** | **56.70%** |

These results mirror the CLIP findings and confirm that the spectral reweighting principle underlying MonoSoup is not architecture-specific.

# I. Variance Threshold $R$ Ablation

The entropy-based effective rank removes the need for a manual threshold (Section 4); here we study the fixed-$R$ variant to understand its sensitivity. We sweep $R \in \{0.1, 0.2, \ldots, 0.9\}$ on three checkpoints: two CLIP ViT-B/32 models fine-tuned from zero-shot initialization (Figure 10) and a ConvNeXt-Small model pretrained on ImageNet-22k and fine-tuned on ImageNet-1k (Figure 11).[2]

**CLIP (zero-shot init.).** Both checkpoints exhibit the same pattern (Figure 10): OOD accuracy peaks at moderate thresholds ($R \approx 0.4$–$0.6$, reaching $\sim 53.3\%$) and declines toward $R=0.9$, while ID accuracy remains flat for $R \leq 0.7$ and rises only at high thresholds ($\sim 79\%$ at $R=0.9$). Interpreting $R$ as the fraction of spectral energy assigned to the high-energy subspace, moderate truncation retains low-energy residuals that benefit OOD, whereas large $R$ steers the model toward the fine-tuned solution at the expense of OOD robustness. The consistency across two independently fine-tuned checkpoints confirms that this trade-off is a property of the spectral structure, not an artifact of a particular training run.

---

[2]Checkpoint: https://huggingface.co/timm/convnext_small.in12k_ft_in1k

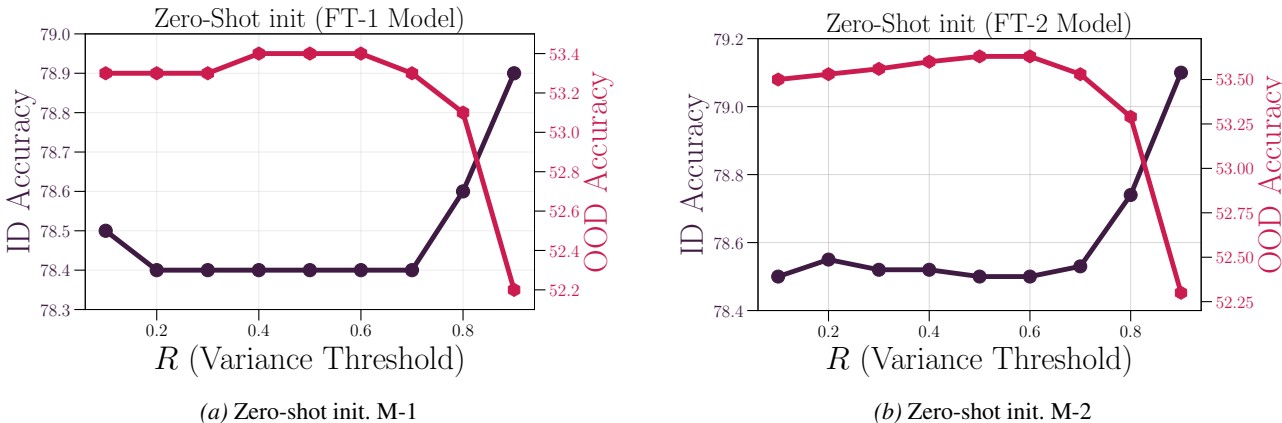

*(a)* Zero-shot init. M-1
                                    *(b)* Zero-shot init. M-2

*Figure 10.* ID and OOD accuracy of MonoSoup on two CLIP ViT-B/32 checkpoints (zero-shot init.) as a function of the variance threshold $R$. Mid-range values ($R \approx 0.5$–$0.7$) favour OOD; large values ($R \geq 0.85$) favour ID.

**ConvNeXt.** The sensitivity is gentler and largely monotonic (Figure 11): both ID and OOD accuracy increase steadily with $R$, with a modest knee near $R \approx 0.7$–$0.8$. This suggests that the ConvNeXt fine-tuning update is spectrally more concentrated than that of CLIP, so expanding the retained subspace recovers useful signal without discarding low-energy components that disproportionately help OOD.

**Takeaway.** Across all three models, $R \in [0.7, 0.85]$ delivers the strongest joint ID–OOD balance. Nevertheless, the automated entropy-based variant matches or exceeds this range without manual selection (Table 5), confirming that $R$ need not be tuned in practice.

## J. Anisotropic Scaling

MonoSoup applies genuinely anisotropic scaling across layers, rather than collapsing to the scalar interpolation of methods such as Wise-FT (Wortsman et al., 2022b), which enforce a uniform scaling factor ($\lambda_{\text{High}}^{(\ell)} = \lambda_{\text{Low}}^{(\ell)} = \alpha$) across all directions. As shown in Figure 12, the mixing coefficients $\lambda_{\text{High}}^{(\ell)}$ and $\lambda_{\text{Low}}^{(\ell)}$ exhibit a distinct and significant separation across layers of the ConvNeXt model, with a mean gap of approximately $0.35$ and $\lambda_{\text{High}}^{(\ell)}$ consistently dominating $\lambda_{\text{Low}}^{(\ell)}$. This allows MonoSoup to preserve the principal task-specific subspace while selectively damping the low-energy orthogonal complement. Furthermore, the magnitude of this separation is *not static*: as seen in the deeper layers (Stage 3) of Figure 12, the gap fluctuates significantly more than in earlier stages (Stage 2), demonstrating that MonoSoup actively adapts to the heterogeneous spectral properties of each layer — a layer-specific geometric transformation that is fundamentally not achievable via scalar interpolation or fixed manual hyperparameters.

## K. Out-of-Distribution Evaluation

The ID/OOD split used in subsection 5.2 (GSM8K $\rightarrow$ GSM$_{\text{Plus}}$/GSM8K$_{\text{Platinum}}$) varies task *difficulty* rather than the underlying distribution. To test whether the gains of MonoSoup extend to genuine distributional shifts, we evaluate on two complementary axes: a *cross-lingual* shift, where the input language differs from the fine-tuning corpus, and a *cross-domain* shift, where the target capability is absent from the fine-tuning data entirely.

**Setup.** Starting from the Qwen3-0.6B base model (Yang et al., 2025) and a variant fine-tuned exclusively on English mathematical data (MetaMathQA (Yu et al., 2024))[3], we apply MonoSoup post-hoc and evaluate all three checkpoints, namely the pretrained, fine-tuned, and MonoSoup models, under identical conditions with `lm-evaluation-harness` (v0.4.11) (Gao et al., 2024).[4] For the cross-lingual benchmarks we use the `mgsm_native_cot_*` tasks with 8-shot native chain-of-thought prompting and the generation parameters recommended by the Qwen3 technical report (Yang et al., 2025) (temperature $= 0.7$, top-$p = 0.8$, top-$k = 20$, seed $= 42$). Under

---

[3]Fine-tuned checkpoint: https://huggingface.co/suayptalha/Qwen3-0.6B-Math-Expert
[4]https://github.com/EleutherAI/lm-evaluation-harness

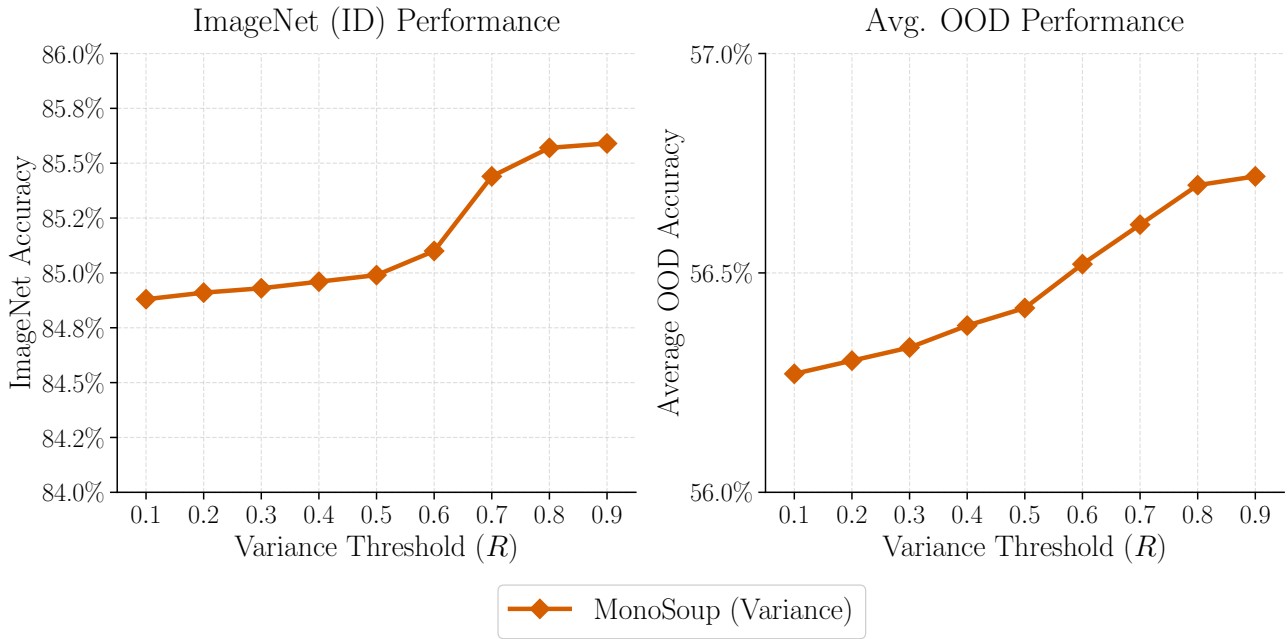

*Figure 11.* Variance threshold ablation on ConvNeXt-Small. Unlike CLIP, both ID and OOD improve monotonically with $R$.

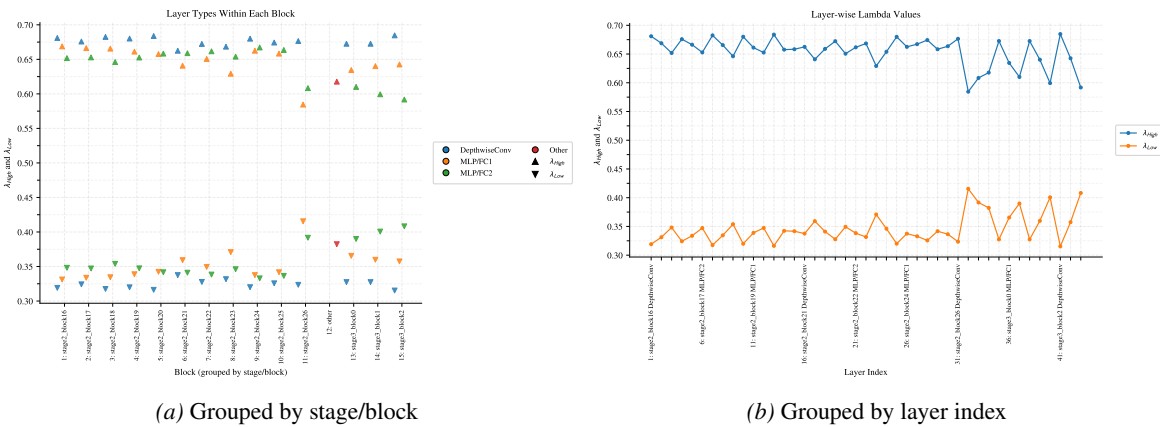

*(a)* Grouped by stage/block        *(b)* Grouped by layer index

*Figure 12.* Evidence of Anisotropic Scaling. Distribution of the computed mixing coefficients $\lambda_{\text{High}}^{(\ell)}$ and $\lambda_{\text{Low}}^{(\ell)}$ across different layers of the ConvNeXt model.

this protocol our pretrained baseline achieves 32.6% average MGSM accuracy, matching the 33.1% reported by Yang et al. (2025) within standard error. Cross-domain benchmarks use their standard 0-shot harness configurations (Clark et al., 2018; Zellers et al., 2019; Hendrycks et al., 2021b).

**Cross-lingual OOD.** Because the fine-tuned model has seen only English, the non-English subsets of MGSM (Shi et al., 2023) isolate language as the sole axis of distributional shift while holding problem difficulty constant. As Table 6 shows, fine-tuning degrades performance in $6/8$ languages (avg. $-2.0$ pp), confirming that task-specific adaptation erodes the multilingual representations acquired during pre-training. MonoSoup reverses this collapse: it improves $7/8$ languages over the fine-tuned checkpoint ($+3.3$ pp on average) and surpasses even the pretrained baseline by $+1.2$ pp, indicating that the spectral reweighting not only restores but *refines* the retained multilingual signal.

**Cross-domain OOD.** We further test on benchmarks whose target capabilities, namely scientific reasoning (ARC-CHALLENGE (Clark et al., 2018)), commonsense inference (HELLASWAG (Zellers et al., 2019)), and humanities/social-science knowledge (MMLU (Hendrycks et al., 2021b)), are entirely absent from the mathematics-focused training data. Table 6 shows that MonoSoup preserves or improves every cross-domain benchmark, gaining $+2$–$3$ pp on the MMLU

*Table 6.* Out-of-distribution evaluation along two axes: cross-lingual shift (top; MGSM, 8-shot native CoT) and cross-domain shift (bottom; 0-shot). $\Delta_{\text{FT}\to\text{Mono}}$ denotes the gain of MonoSoup over the fine-tuned checkpoint. **Bold** marks the best score per row.

| | Pretrained | Fine-tuned | MonoSoup | $\Delta_{\text{FT}\to\text{Mono}}$ |
|---|---|---|---|---|
| *Cross-lingual OOD — MGSM (8-shot CoT)* | | | | |
| EN | 45.2 | 40.4 | **47.6** | +7.2 |
| ES | 35.2 | 33.6 | **37.1** | +3.5 |
| FR | 35.6 | 33.6 | **37.5** | +3.9 |
| DE | 28.8 | 24.8 | **30.2** | +5.4 |
| ZH | 40.0 | 39.2 | 38.8 | −0.4 |
| JA | 19.2 | 20.8 | **21.1** | +0.3 |
| RU | 33.6 | 31.6 | **34.8** | +3.2 |
| TH | 22.8 | 20.4 | **23.4** | +3.0 |
| **Avg.** | 32.6 | 30.6 | **33.8** | +3.3 |
| *Cross-domain OOD — general (0-shot)* | | | | |
| GSM8K (ID) | 42.08 | 44.81 | **48.87** | +4.06 |
| ARC-Challenge | 31.48 | 32.94 | **33.11** | +0.17 |
| HellaSwag | 37.43 | 37.92 | 37.85 | −0.07 |
| MMLU Humanities | 36.66 | 39.77 | **41.91** | +2.14 |
| MMLU Social Sci. | 47.55 | 52.52 | **55.36** | +2.84 |

subsets while simultaneously boosting the in-distribution GSM8K score by $+4.06\,\text{pp}$. This confirms that MonoSoup does not trade cross-domain breadth for task-specific depth.

**Discussion.** The cross-lingual and cross-domain results together demonstrate that the benefits of MonoSoup are not confined to difficulty perturbations of the training distribution but extend to bona fide distributional shifts along both linguistic and capability axes.

**Computational Cost.** MonoSoup is a *one-shot, data-free* spectral operation applied post hoc to a single fine-tuned checkpoint: it requires neither gradient computation nor access to training data. We profile its wall-clock time, peak GPU memory, and storage footprint across four model scales spanning more than two orders of magnitude in parameter count, and contrast against the multi-checkpoint cost of Model Soups (Wortsman et al., 2022a). All profiling is performed on a single NVIDIA A100 GPU (80 GB). For each model we load the pretrained and fine-tuned state dicts, compute the layer-wise weight difference $\Delta\boldsymbol{W}^{(\ell)}$, and apply the full MonoSoup pipeline (SVD, entropy-based effective rank, coefficient computation, reweighting) sequentially across all 2-D weight matrices; we report the median wall-clock time over three runs. For the CLIP models we use the ViT-B/32 and ViT-L/14 checkpoints released by Wortsman et al. (2022a), fine-tuned on ImageNet with linear-probing initialization. For Qwen3-0.6B we pair the base model (Yang et al., 2025) with a variant fine-tuned on MetaMathQA (Yu et al., 2024).[5] For Qwen3-14B we pair the base model with Ophiuchi-Qwen3-14B-Instruct,[6] an instruction-tuned variant trained on mathematical reasoning and code generation data, ensuring that the SVD operates on realistic fine-tuning deltas.

On the primary CLIP setting MonoSoup completes in $2-3$ seconds; even on Qwen3-14B it takes only $\sim 83$ seconds, versus the $\sim 240$ GPU-hours required to train the ten soup checkpoints. Because the decomposition is performed layer-wise, peak memory scales with the largest individual weight matrix rather than the full model, keeping the procedure tractable on a single commodity GPU. Storage is likewise reduced by an order of magnitude (e.g. 56 GB versus 560 GB for Qwen3-14B, since only one checkpoint must be retained).

## L. Compatibility with Parameter-Efficient Fine-Tuning

**Positioning relative to spectral PEFT methods.** A line of parameter-efficient fine-tuning (PEFT) work also examines the spectral structure of weight updates, and we clarify its relationship to MonoSoup before presenting our experiment. LoRA-Pro (Wang et al., 2025b) operates at *training time*: it adjusts the gradients of the low-rank factors $\boldsymbol{A}, \boldsymbol{B}$ so that the resulting

---

[5] https://huggingface.co/suayptalha/Qwen3-0.6B-Math-Expert
[6] https://huggingface.co/prithivMLmods/Ophiuchi-Qwen3-14B-Instruct

*Table 7.* Computational cost of MonoSoup versus Model Soups across four model scales. MonoSoup is a single-checkpoint, data-free spectral decomposition; the Model Soup column estimates the wall-clock cost of independently fine-tuning $m=10$ checkpoints with different hyperparameter configurations, as required by Wortsman et al. (2022a). Peak memory reflects the largest single layer, since the decomposition is applied layer-wise. [†] Estimate of the total wall-clock time to independently fine-tune ten checkpoints under different hyperparameter configurations, as required by Model Soups. Exact cost depends on dataset size, hardware, and training recipe; the order-of-magnitude gap is robust to all three.

| Model | Params | MonoSoup | Peak Mem | Model Soup[†] | Speedup |
|---|---|---|---|---|---|
| CLIP ViT-B/32 | 88M | 3.1 s | 685 MB | $\sim 150$ min | $\sim 2{,}900\times$ |
| CLIP ViT-L/14 | 304M | 2.2 s | 1,024 MB | $\sim 900$ min | $\sim 25{,}000\times$ |
| Qwen3-0.6B | 600M | 18.8 s | 4,170 MB | $\sim 1{,}200$ min | $\sim 3{,}800\times$ |
| Qwen3-14B | 14B | 83.3 s | 20,884 MB | $\sim 14{,}400$ min | $\sim 10{,}400\times$ |

*Table 8.* Effect of MonoSoup applied to a LoRA-fine-tuned checkpoint (Qwen3-0.6B, $r=8$, RSLoRA, MetaMathQA). **Bold** indicates LoRA + MonoSoup attains the best score for that row.

| Benchmark | LoRA FT | LoRA + MonoSoup | $\Delta$ |
|---|---|---|---|
| GSM8K | 31.92 | **32.60** | +0.68 |
| MGSM EN | 32.40 | **34.80** | +2.40 |
| MGSM DE | 23.60 | **24.80** | +1.20 |
| MGSM ZH | 30.80 | **33.60** | +2.80 |
| MGSM JA | 18.40 | **19.20** | +0.80 |

low-rank update more closely approximates the gradient of full fine-tuning, with approximation fidelity as its objective. Related methods reason about spectra differently: PiSSA (Meng et al., 2024) initialises adapters from the principal singular components of the pretrained weight $W$ (not of the task vector $\Delta W$) and freezes the low-energy residual. MonoSoup differs along three axes: it acts *post hoc* on an already fine-tuned checkpoint rather than during training; its objective is to recover the ID–OOD balance of multi-checkpoint ensembling rather than to approximate a full-rank gradient; and its mechanism is a reweighting of the singular-value spectrum of the accumulated update $\Delta W = W_{\mathrm{FT}} - W_0$, rather than a modification of the learning dynamics. These prior methods analyse low-energy directions in the task-arithmetic or low-rank-adapter regimes; our contribution is the finding that, under large-scale full-rank fine-tuning with natural distribution shifts, the low-energy tail of $\Delta W$ carries information essential for OOD robustness (Figure 3), together with an automated, data-free scheme that exploits it. Crucially, this makes the two approaches *complementary* rather than competing, as the experiment below confirms.

**Applying MonoSoup to a LoRA checkpoint.** To assess whether MonoSoup extends beyond full-rank updates, we fine-tune Qwen3-0.6B (Yang et al., 2025) with LoRA (Hu et al., 2022) on 100k samples from MetaMathQA (Yu et al., 2024) for one epoch using rank $r=8$, RSLoRA scaling (Kalajdzievski, 2023), $\alpha=16$, and no dropout, targeting all attention and MLP projections (`q/k/v/o_proj`, `gate/up/down_proj`). Training uses AdamW with a cosine schedule (learning rate $2\times10^{-4}$, effective batch size 32) on a single A100 GPU. After training we merge the adapter into the base weights to obtain a full-rank checkpoint, then apply MonoSoup to the merged model. Both the LoRA-merged and LoRA + MonoSoup checkpoints are evaluated under the same protocol as Appendix K: 8-shot native CoT for MGSM and the standard harness configuration for GSM8K.

Table 8 shows that MonoSoup improves 5/6 benchmarks, but the absolute gains are attenuated relative to the full-rank regime (MGSM average $+1.8$ pp here vs. $+3.3$ pp in Table 6). This attenuation is predicted by the structure of the update. A LoRA update factorises as $\Delta W = BA^\top$ with $A, B \in \mathbb{R}^{d\times r}$, so $\mathrm{rank}(\Delta W) \leq r$ and the singular spectrum of $\Delta W$ satisfies $\sigma_i(\Delta W) = 0$ for all $i > r$. Since MonoSoup acts by attenuating the trailing singular components of $\Delta W$, and that tail is identically zero beyond rank $r$, the operator's intervention surface is at most $r$-dimensional. MonoSoup is therefore most effective when applied to full-rank updates, where the spectral tail carries informative mass; in the strictly rank-bounded LoRA regime it still yields consistent but quantitatively smaller improvements, as Table 8 confirms. This also explains why MonoSoup and training-time spectral methods such as LoRA-Pro are complementary: a spectrum-aware PEFT method and the post-hoc MonoSoup step act at different stages of the pipeline and on different objects, so one may apply MonoSoup to a LoRA-Pro-trained checkpoint after merging, with the post-hoc gain governed by whatever informative mass remains in the low-energy tail.

*Table 9.* **Six pairs with extreme negative outliers.** For each pair we report the individual ID and OOD accuracies of the two constituent checkpoints, alongside the ID and OOD accuracy of their Model Stock combination.

| Model Pair | | ID Acc. (%) | | OOD Acc. (%) | | Model Stock | |
|---|---|---|---|---|---|---|---|
| Model $i$ | Model $j$ | $\text{Acc}_{\text{ID}}^i$ | $\text{Acc}_{\text{ID}}^j$ | $\text{Acc}_{\text{OOD}}^i$ | $\text{Acc}_{\text{OOD}}^j$ | ID | OOD |
| Model-2 | Model-14 | 76.2 | 76.5 | 37.1 | 36.7 | 13.2 | 6.0 |
| Model-1 | Model-10 | 77.2 | 78.4 | 47.4 | 48.4 | 72.5 | 42.5 |
| Model-11 | Model-12 | 77.3 | 77.3 | 44.2 | 47.2 | 75.5 | 46.8 |
| Model-42 | Model-43 | 78.7 | 80.3 | 41.5 | 47.9 | 30.4 | 12.7 |
| Model-54 | Model-55 | 79.2 | 76.4 | 47.5 | 43.5 | 46.7 | 22.4 |

