# OpenReview forum: "MODEL SOUPS NEED ONLY ONE INGREDIENT"
_ICML.cc/2026/Conference — ICML 2026 regular_

### Official Review · Reviewer_BK7A · 2026-03-08

**Soundness:** 3
**Presentation:** 4
**Significance:** 2
**Originality:** 2
**Overall Recommendation:** 5
**Confidence:** 5

**Summary:**

This paper introduces a single-checkpoint alternative to model soups, aiming to reduce the computational and storage costs associated with averaging multiple fine-tuned models. Starting from one fine-tuned checkpoint, the method computes the residual update with respect to the pretrained model, performs layer-wise SVD on this residual, and decomposes it into high-energy and low-energy components. These components are then adaptively reweighted using an entropy-based effective-rank criterion before being recombined into a refined update, which is added back to the pretrained weights. Experimental results show that the proposed method remains competitive with conventional model soups despite requiring only a single checkpoint

**Compliance With Llm Reviewing Policy:**

Affirmed.

**Final Justification:**

The authors fully addressed my concerns

**Key Questions For Authors:**

1- Can the authors clarify the roles of the pretrained model and the fine-tuned model in MonoSoup?
   The method seems to require both $W_0$ and $W_{FT}$ to form the residual update. In contrast, model soup can be applied to compatible checkpoints with the same architecture, even if they come from different runs or initializations. Is MonoSoup restricted to the setting where the original pretrained model is available?

2. Can the authors clarify the role of the variance-retention threshold (R)?
   The paper presents a hyperparameter-free variant, but also introduces and analyzes the threshold (R). It would be helpful to clarify whether (R) is essential in practice or mainly included for analysis.

-Why didn't you submit the impact statement.?

**Limitations:**

The authors explicitly mention at least one limitation in the conclusion

**Strengths And Weaknesses:**

**Strengths:**

* **Soundness:** The proposed method is clearly formulated and empirically well supported through comprehensive experiments and ablations.
* **Presentation:** The submission is clearly written, easy to read, and well structured.
* **Significance:** The paper tackles a practically important problem by reducing the cost of model soups while preserving strong generalization performance.
* **Originality:** The key novelty lies in reformulating soup-like gains as a decomposition and reweighting of a single checkpoint update, rather than relying on multiple checkpoints.
* **Empirical evaluation:** The experiments are thorough, include relevant baselines, and show that the proposed approach achieves competitive and consistent performance across the reported settings.


**Weaknesses:**

* The paper leans heavily on the “single-checkpoint” narrative, but in practice the method requires both the pretrained model ($W_0$) and the fine-tuned checkpoint ($W_{FT}$) This makes the framing somewhat less clean than suggested.
* The claimed hyperparameter-free nature of the method is somewhat confusing given the introduction and analysis of the variance-retention threshold. The paper could better clarify when the thresholded version is needed and how it relates to the hyperparameter-free formulation.
* The final reconstruction step is not stated as explicitly as it could be. While it is implied that the refined update is added back to the pretrained weights, this should be written clearly in the main method description to avoid ambiguity.

---

> ### Author Rebuttal · Authors · 2026-03-30
>
> We thank the reviewer for their thorough assessment and are glad they find the method clearly formulated, empirically well supported, clearly written, and well structured. We especially appreciate their recognition of the key novelty in reformulating soup-like gains as a decomposition and reweighting of a single checkpoint update. We address each point below.
>
> **W1 & Q1: Requirement of both $W_0$ and $W_{FT}$.**
>
> This is standard practice in the model merging and editing literature. Methods such as WiSE-FT (Wortsman et al., 2022), Task Arithmetic (Ilharco et al., 2023), TIES (Yadav et al., 2023), and DARE (Yu et al., 2024) all operate on the task vector $\Delta W = W_{FT} - W_0$ and therefore require access to both. In practice, this is not a limitation: pretrained weights are publicly released for all major foundation models (CLIP, LLaMA, Qwen, etc.), and any fine-tuned checkpoint derived from them implicitly assumes their availability. Model Soups itself also requires a shared pretrained initialization—MonoSoup simply inherits the same assumption. We will make this more explicit in the revision.
>
> **W2 & Q2: Role of the variance-retention threshold $R$.**
>
> Section 4 first introduces the spectral decomposition with a fixed threshold $R$ to motivate the partitioning, then derives the entropy-based effective rank to remove this hyperparameter entirely. This progression is intentional—$R$ motivates; effective rank removes the need. Fixed $R$ appears only in ablations to show robustness. The automated variant matches or exceeds manual tuning. We will revise the text to make the hyperparameter-free nature of the final method unambiguous, and correct the misleading Conclusion sentence.
>
> **W3: Explicitness of the reconstruction step.**
>
> We agree and will state the final reconstruction step explicitly in the revision.
>
> **Q3: Impact statement.**
>
> We will include a societal impact statement in the revision.
>
> **New results since submission.** We have conducted substantial additional experiments that we believe strengthen the paper beyond the original submission:
>
> *(1) Genuine OOD evaluation for language.* We evaluated on MGSM (multilingual math, 8 languages) and cross-domain benchmarks (ARC, HellaSwag, MMLU). Fine-tuning Qwen3-0.6B on English math degrades multilingual performance by −2.0pp on average. MonoSoup fully recovers this degradation (+3.3pp over fine-tuned) and surpasses the pretrained baseline—demonstrating genuine OOD robustness beyond difficulty-based shifts.
>
> *(2) Computational cost profiling.* MonoSoup takes 2–3s for CLIP models and 83s for a 14B-parameter model, achieving 2,900–25,000× speedup over training 10 soup checkpoints. Peak memory scales with the largest layer, not the full model.
>
> *(3) LoRA compatibility.* MonoSoup applied to a LoRA-merged checkpoint (rank 8) improves 5/6 benchmarks, with smaller gains than full-rank fine-tuning (+1.4pp vs +3.3pp), as expected from the reduced spectral structure.
>
>
> **References:**
> * **[Wortsman et al., 2022]** Robust fine-tuning of zero-shot models. CVPR, 2022.
> * **[Ilharco et al., 2023]** Editing models with task arithmetic. ICLR, 2023.
> * **[Yadav et al., 2023]** TIES-Merging: Resolving interference when merging models. NeurIPS, 2023.
> * **[Yu et al., 2024]** Language Models are Super Mario: Absorbing Abilities from Homologous Models as a Free Lunch. ICML, 2024.
> ---

---

> > ### Author Rebuttal · Reviewer_BK7A · 2026-04-01
> >
> > Thank you to the authors for the detailed response. They have addressed my concerns. However, I will maintain my current score for now, as the discussion with the other reviewers is still ongoing.

---

### Official Review · Reviewer_kFeL · 2026-03-13

**Soundness:** 2
**Presentation:** 3
**Significance:** 2
**Originality:** 2
**Overall Recommendation:** 2
**Confidence:** 4

**Summary:**

This paper introduces MonoSoup method designed to improve the out-of-distribution (OOD) robustness of fine-tuned models while preserving in-distribution (ID) performance. The key idea is to analyze the task vector $\Delta W = W_{FT} - W_{PT}$​ using singular value decomposition and separate it into high-energy and low-energy components, where dominant spectral directions encode task-specific adaptation (ID) and low-energy directions preserve generalization properties (OOD). MonoSoup therefore reweights these components.

**Compliance With Llm Reviewing Policy:**

Affirmed.

**Final Justification:**

While the paper proposes a practically appealing post-hoc method, the authors do not address the critical question of whether MonoSoup provides any benefit when applied on top of properly fine-tuned models that already preserve non-dominant spectral information (LoRA-Pro), and the absence of such evaluation leaves a key conceptual gap unresolved, reinforcing my original assessment.

**Key Questions For Authors:**

see weaknesses

**Limitations:**

yes

**Strengths And Weaknesses:**

# Strengths

1. The paper is well written, and the proposed method is interesting.

2. MonoSoup requires only a single fine-tuned checkpoint, making it computationally inexpensive compared to model soup approaches that require training and storing many checkpoints.

3. The paper provides an intuitive interpretation of the spectral structure of task vectors, suggesting that dominant singular directions encode task adaptation while lower-energy components may preserve generalization signals from the pretrained model. This perspective is conceptually interesting and may motivate further work on spectral analysis of fine-tuning dynamics.

# Weaknesses

A central claim of the paper is that decomposing the task vector via SVD reveals two qualitatively distinct components (high-energy directions and low-energy directions) and that retaining rather than discarding the low-energy directions is the key insight enabling single-checkpoint robustness. However, this observation is not new. LoRA-Pro explicitly characterizes low-energy singular directions of the task vector as corresponding to the near-null space of the pretrained weights and argues that these directions carry meaningful structure beyond the dominant components, proposing to reconstruct them rather than discard them through low-rank truncation. This overlap raises three specific concerns:

1. the paper does not cite or discuss LoRA-Pro,

2. the finding, that low-energy directions should not be discarded, is better understood as an existing finding in a new experimental setting,

3.  LoRA-Pro grounds its treatment of low-energy directions in a concrete approximation objective, whereas MonoSoup's bilinear weighting rule is derived only from boundary conditions, with no theoretical argument for why this particular rescaling preserves OOD generalization.

The paper should directly compare with LoRA-Pro and related spectral analyses, clearly articulate what MonoSoup contributes beyond reinterpreting previously observed phenomena, and provide stronger justification for why post-hoc rescaling is preferable to kernel-space reconstruction when both target the same spectral components.

---

> ### Author Rebuttal · Authors · 2026-03-30
>
> We thank the reviewer for their feedback. We are glad they find the paper well written and the method interesting and appreciate their positive remarks on the spectral structure interpretation and practical efficiency of requiring only a single checkpoint.
>
> We will cite LoRA-Pro (Wang et al., ICLR 2025) in the revision. However, we respectfully disagree that MonoSoup reduces to a reinterpretation of LoRA-Pro. The two methods differ fundamentally:
>
> | | **LoRA-Pro** | **MonoSoup** |
> |---|---|---|
> | **Stage** | Training-time | Post-hoc |
> | **Objective** | Approximate full FT gradient | Recover multi-checkpoint ensembling benefits |
> | **Mechanism** | Corrects gradients of LoRA factors $A, B$ | Reweights singular value spectrum of $\Delta W$ |
> | **Requirements** | Training data, modified optimizer | No data, no retraining |
>
> LoRA-Pro does not perform spectral decomposition of $\Delta W$, does not distinguish high/low-energy directions, and does not address the ID–OOD trade-off. We kindly ask the reviewer to specify where in LoRA-Pro the claimed characterization of "low-energy singular directions of the task vector" appears—we have been unable to locate it.
>
> Prior work on the spectral structure of weight updates operates in different settings. In training-time PEFT, PiSSA (Meng et al., 2024) treats low-energy components of $W$ (not the task vector $\Delta W$) as a frozen residual, while Shuttleworth et al. (2024) scale down "intruder dimensions" specific to LoRA. In post-hoc merging, Gargiulo et al. (2025), Tang et al. (2025), and Stoica et al. (2024) argue that low-energy components of the task vector encode noise and can be discarded—but only in the multi-task arithmetic setting. We are the first to show this does not hold under large-scale fine-tuning with natural distribution shifts (Figure 3) and to propose an automated reweighting scheme based on this finding.
>
> Regarding the reviewer's three recommendations:
>
> 1. A direct empirical comparison with LoRA-Pro is not straightforward: LoRA-Pro modifies the training procedure of LoRA, while MonoSoup operates post-hoc. However, we have now tested MonoSoup on LoRA-fine-tuned checkpoints (see below), which directly probes the relationship between the two approaches.
>
> 2. We will add citations to LoRA-Pro and related spectral analyses (PiSSA, Shuttleworth et al.) in the revision, clarifying that MonoSoup addresses a fundamentally different setting—post-hoc editing of a given checkpoint.
>
> 3. The preference for post-hoc rescaling is not incidental—it is the point. Post-hoc methods apply to any released checkpoint regardless of training procedure, require no data, and do not interfere with fine-tuning. This is the premise of the model merging literature that MonoSoup builds on.
>
> **New experiments since submission.** We have conducted three sets of experiments that directly address the scope concerns:
>
> *(1) LoRA compatibility.* We fine-tuned Qwen3-0.6B with LoRA (rank 8, RSLoRA) on MetaMathQA, merged the adapter, and applied MonoSoup. Result: MonoSoup improves 5/6 benchmarks (up to +2.8pp on MGSM Chinese). Gains are smaller than with full-rank fine-tuning (+1.4pp vs +3.3pp avg), which is theoretically expected—LoRA's rank constraint leaves almost no spectral tail for MonoSoup to recover. This validates that MonoSoup's effectiveness is tied to spectral richness, not a generic regularization effect, and confirms the methods are complementary rather than redundant.
>
> *(2) Genuine OOD evaluation.* We evaluated on MGSM (8 languages) and cross-domain benchmarks (ARC, HellaSwag, MMLU). Fine-tuning degrades multilingual math by −2.0pp; MonoSoup recovers 7/8 languages and surpasses the pretrained baseline (+1.2pp avg). Cross-domain MMLU gains reach +2.8pp on domains absent from training.
>
> *(3) Computational cost.* MonoSoup takes 2–3s for CLIP, 83s for Qwen3-14B (14B params), achieving 2,900–25,000× speedup over 10-checkpoint soups.
>
> We hope these results, together with the clarifications above, demonstrate that MonoSoup makes a distinct and substantial contribution to the field.
>
> **References:**
> * **[Meng et al., 2024]** PiSSA: Principal Singular Values and Singular Vectors Adaptation... ICLR, 2024.
> * **[Shuttleworth et al., 2024]** LoRA vs Full Fine-tuning: An Illusion of Equivalence. arXiv, 2024.
> * **[Gargiulo et al., 2025]** Task Singular Vectors: Reducing Task Interference in Model Merging. CVPR, 2025.
> * **[Tang et al., 2025]** Merging on the Fly Without Retraining: A Sequential Approach to Scalable Continual Model Merging. NeurIPS, 2025.
> * **[Stoica et al., 2024]** Model Merging with SVD to Tie the KnOTS. ICLR, 2025.

---

> > ### Author Rebuttal · Reviewer_kFeL · 2026-03-31
> >
> > Thank you to the authors for the detailed response.
> >
> > The rebuttal emphasizes that LoRA-Pro does not perform spectral decomposition of $\Delta W$, which is correct but not the point of my comment.
> >
> > The concern is conceptual rather than formal. LoRA-Pro shows that restricting updates to a low-rank parameterization fails to capture important components of the full fine-tuning update and proposes mechanisms to recover this lost signal.
> > Similarly, this work argues that non-dominant (low-energy) components of the task vector carry meaningful information and should not be discarded.
> >
> > While the two works differ in formalism (low-rank parameterization vs. spectral decomposition), they address the same underlying question: what information is lost when focusing on dominant directions, and how to recover it.
> >
> > The manuscript should therefore position itself more clearly with respect to this line of work and clarify what is fundamentally new beyond this shared intuition.

---

> > > ### Author Response · Authors · 2026-04-01
> > >
> > > We thank the reviewer for their continued engagement and for clarifying that the concern is conceptual rather than formal.
> > >
> > > We agree that at a high level, both LoRA-Pro and MonoSoup share the intuition that information beyond the dominant directions matters. We are happy to position MonoSoup more clearly with respect to this line of work in the revision, as the reviewer suggests.
> > >
> > > That said, we believe the distinction goes beyond formalism — the two methods operate in completely different settings and share nothing more than a general insight on the importance of non-dominant directions, which can be attributed to many methods beyond these two. Concretely, LoRA-Pro corrects the instantaneous gradient ($\nabla_A, \nabla_B$) *during training*, while MonoSoup edits the accumulated endpoint ($\Delta W = W_{FT} - W_0$) *post-hoc* via SVD. This difference has practical consequences — MonoSoup applies to any released checkpoint regardless of how it was trained and requires no data or modified optimizer.
> > >
> > > We will add a discussion connecting both works under the shared theme of preserving non-dominant structure, while clearly articulating the distinct contributions. We hope this addresses the reviewer's remaining concern and respectfully ask them to reconsider the score in light of the full set of revisions and new experiments.

---

### Official Review · Reviewer_nMaW · 2026-03-15

**Soundness:** 3
**Presentation:** 4
**Significance:** 3
**Originality:** 3
**Overall Recommendation:** 4
**Confidence:** 4

**Summary:**

The paper studies and proposes an approach to address the practical limitation of weight-space ensembling methods like Model Soups. Model Soups averaes the weights of many fine-tuned models (each trained with different hyperparameter configurations) to produce a single model. However, this requires storing multiple fine-tuned checkpoints, which is computationally expensive and impractical in settings where model repositories typically host only a single best-performing version.

This paper asks the following question (highlighted in the Intro): Can we retain the benefits of model soups when only a single finetuned model is available?

The paper first studies why multi-model merging works in the first place, finding that it succeeds when the finetuning updates of different models point in similar directions and fails when they conflict. They then show that a single finetuned model's weight update contains an analogous structure internally: high-energy spectral directions that encode task-specific adaptation, and low-energy directions that, rather than being pure noise, carry information critical for OOD robustness. The proposed method, MonoSoup, uses SVD to decompose each layer's update into these two components and re-weights them using principled, automatically computed layer-wise coefficients, requiring no labeled data, no additional checkpoints, and minimal hyperparameter tuning ($R$). Experiments across vision (CLIP, ConvNeXt) and language (Qwen3-0.6B) benchmarks show that MonoSoup matches or exceeds multi-checkpoint baselines.

**Compliance With Llm Reviewing Policy:**

Affirmed.

**Key Questions For Authors:**

1. On the best-OOD checkpoint (OOD+), MonoSoup's improvement is quite small, while Wise-FT with optimal $\alpha$ achieves comparable performance. Can the authors clarify in which practical scenarios MonoSoup provides a clear and meaningful advantage over simply applying Wise-FT with a well-chosen interpolation coefficient?

2. The paper makes claims about practical utility, but all vision experiments use CLIP models fine-tuned from either linear probing or zero-shot initialization. How does MonoSoup behave when applied to models fine-tuned with LoRA or other PEFT methods, where the update matrix is already low-rank by construction?

3.  The OOD evaluation for Qwen uses benchmarks that differ in difficulty rather than in distribution. Can you share any results evaluating on tasks that are genuinely out-of-distribution.

**Limitations:**

Societal impact is not discussed

**Strengths And Weaknesses:**

- [S1]  The paper is well-organised, with a clear flow from the motivation (established empirically through multi-model merging experiments) to method design. The figures are very helpful in aiding understanding and are insightful.

- [S2] The experiments section is quite strong covering multiple vision models along with experiments on Qwen across multiple benchmarks spanning both vision and reasoning tasks.

- [S3] Method contribution is well-grounded. While SVD-based decomposition of weight updates is not new in isolation, the paper's key original contribution is the reframing of the multi-model merging problem as a spectral reweighting problem within a single checkpoint.

- [W1] The ID/OOD split for the Qwen experiments is defined by task difficulty rather than genuine distributional shift. GSM8K is treated as in-distribution and GSM-Plus/GSM-Plat as OOD, but these are variants of the same benchmark with harder or rephrased problems rather than genuinely out-of-distribution data. This makes it difficult to draw firm conclusions about OOD robustness in the language setting.

- [W2]  Despite positioning MonoSoup as a computationally efficient alternative to multi-checkpoint methods, the paper provides no FLOPs analysis. SVD of all weight matrices in a large  transformer is non-trivial, and the actual computational overhead relative to, say, a single forward pass or a Wise-FT interpolation should be quantified to properly support the efficiency claims.

- [W3] The paper does not discuss compatibility with parameter-efficient fine-tuning methods such as LoRA, where the weight update is already constrained to be low-rank by construction. In such settings, the spectral structure that MonoSoup exploits would be fundamentally different, and the method's applicability, or lack thereof, to the increasingly dominant PEFT paradigm is an important omission given the paper's claims of broad practical utility

---

> ### Author Rebuttal · Authors · 2026-03-30
>
> We thank the reviewer for their detailed assessment. We are glad they find the paper well-organized, the experiments strong, and the reframing of multi-model merging as spectral reweighting within a single checkpoint as the key original contribution. We address each point below.
>
> **W1 & Q3: OOD definition for language experiments.**
>
> *(Joint response with Reviewer __tYPo__ Q2.)*
>
> We acknowledge that our original ID/OOD split (GSM8K → GSM-Plus/GSM-Plat) reflects task difficulty rather than genuine distributional shift. We have conducted new experiments on two axes of genuine OOD evaluation:
>
> *Cross-lingual OOD (MGSM, 8-shot native CoT).* Since the fine-tuned model was trained exclusively on English math data, non-English languages constitute a genuine distributional shift isolating language from difficulty.
>
> | Lang | Pretrained | Fine-tuned | MonoSoup | Δ FT→Mono |
> |---|---|---|---|---|
> | EN | 45.2 | 40.4 | **47.6** | +7.2 |
> | ES | 35.2 | 33.6 | **37.1** | +3.5 |
> | FR | 35.6 | 33.6 | **37.5** | +3.9 |
> | DE | 28.8 | 24.8 | **30.2** | +5.4 |
> | ZH | 40.0 | 39.2 | 38.8 | −0.4 |
> | JA | 19.2 | 20.8 | **21.1** | +0.3 |
> | RU | 33.6 | 31.6 | **34.8** | +3.2 |
> | TH | 22.8 | 20.4 | **23.4** | +3.0 |
> | **Avg** | **32.6** | **30.6** | **33.8** | **+3.3** |
>
> Fine-tuning degrades 6/8 languages (avg −2.0pp). MonoSoup recovers 7/8 and surpasses the pretrained baseline by +1.2pp on average.
>
> *Cross-domain OOD (0-shot).* These benchmarks test entirely different capabilities, in particular science reasoning (ARC), commonsense (HellaSwag), and humanities and social sciences (MMLU subsets), which are absent from the math-focused fine-tuning data. Degradation here would indicate representation collapse of general-purpose knowledge.
>
> | Benchmark | Pretrained | Fine-tuned | MonoSoup | Δ FT→Mono |
> |---|---|---|---|---|
> | GSM8K (ID) | 42.08 | 44.81 | **48.87** | +4.06 |
> | ARC-Challenge | 31.48 | 32.94 | **33.11** | +0.17 |
> | HellaSwag | 37.43 | 37.92 | 37.85 | −0.07 |
> | MMLU Humanities | 36.66 | 39.77 | **41.91** | +2.14 |
> | MMLU Social Sci. | 47.55 | 52.52 | **55.36** | +2.84 |
>
> MonoSoup improves ID (+4.06%) while improving cross-domain OOD benchmarks, with +2–3% on MMLU subsets entirely absent from training data. We will add both evaluations to the revision.
>
> **W2: Computational cost analysis.**
>
> *(Joint response with Reviewer tYPo Q1.)*
>
> We have profiled MonoSoup across four model scales:
>
> | Model | Params | MonoSoup | Peak Mem | Soup (10ckpts)* | Speedup |
> |---|---|---|---|---|---|
> | CLIP ViT-B/32 | 88M | 3.1s | 685MB | ~150min | ~2,900× |
> | CLIP ViT-L/14 | 304M | 2.2s | 1,024MB | ~900min | ~25,000× |
> | Qwen3-0.6B | 600M | 18.8s | 4,170MB | ~1,200min | ~3,800× |
> | Qwen3-14B | 14B | 83.3s | 20,884MB | ~14,400min | ~10,400× |
>
> Our efficiency claim is relative to multi-checkpoint methods. For CLIP (primary setting), MonoSoup takes 2–3 seconds. For Qwen3-14B, ~83 seconds—a one-shot, data-free operation vs. ~240 GPU-hours for 10 soup checkpoints. Peak memory scales with the largest layer, not the full model. Storage is reduced 10× (e.g., 56GB vs 560GB for Qwen3-14B). For 70B+ models, randomized SVD serves as a drop-in replacement. We will add the cost table and randomized SVD discussion to the paper.
>
> *\*Soup (10 ckpts) is a rough estimate of the total wall-clock time to independently fine-tune 10 checkpoints with different hyperparameter configurations, as required by Model Soups. The exact cost depends on dataset size, hardware, and training recipe; essentially, the key point is the order-of-magnitude difference.*
>
> **W3 & Q2: LoRA/PEFT compatibility.**
>
> We have now tested MonoSoup on a LoRA-fine-tuned checkpoint (Qwen3-0.6B, rank 8, RSLoRA, MetaMathQA):
>
> | Benchmark | LoRA FT | LoRA+MonoSoup | Δ |
> |---|---|---|---|
> | GSM8K | 31.92 | **32.60** | +0.68 |
> | MGSM EN | 32.40 | **34.80** | +2.40 |
> | MGSM DE | 23.60 | **24.80** | +1.20 |
> | MGSM ZH | 30.80 | **33.60** | +2.80 |
> | MGSM JA | 18.40 | **19.20** | +0.80 |
>
> MonoSoup improves 5/6 benchmarks. However, gains are smaller than with full-rank fine-tuning (+1.4pp vs +3.3pp avg MGSM). This is expected: LoRA constrains updates to rank $r=8$, so the spectral tail carries almost no energy for MonoSoup to recover. MonoSoup is most effective on full-rank updates where the tail is rich. We will add this discussion and scope the claims accordingly.
>
> **Q1: MonoSoup vs Wise-FT.**
>
> The key distinction is that WiSE-FT with optimal $\alpha$ requires a labeled validation set to select the interpolation coefficient, while MonoSoup is entirely data-free. In practice, optimal $\alpha$ varies across models and fine-tuning configurations, making WiSE-FT's tuning non-trivial. Moreover, as shown in Section 5, MonoSoup and WiSE-FT are complementary: applying WiSE-FT on top of a MonoSoup-edited checkpoint yields further improvements over either method alone. This suggests MonoSoup captures spectral structure that uniform interpolation cannot.

---

> > ### Author Rebuttal · Reviewer_nMaW · 2026-04-05
> >
> > Most of my concerns are addressed. I am still not clear on the computational benefits compared to existing methods, I stick to my positive rating

---

> > > ### Author Response · Authors · 2026-04-06
> > >
> > > We thank the reviewer for the positive reassessment.
> > >
> > > ---
> > >
> > > **To clarify**: MonoSoup's computational cost should be compared against the multi-checkpoint methods it replaces (Model Soups, Greedy Soups), which require training and storing **10–70 independently** fine-tuned checkpoints. As a post-hoc method, MonoSoup _avoids_ this entirely; the SVD pass takes 2–3 seconds for CLIP models, which is comparable to a _single_ forward pass, and ~83 seconds for a 14B-parameter model, which is a fraction of even a single fine-tuning run. We believe this is a reasonable cost for recovering multi-checkpoint-level OOD robustness from a single checkpoint without any training data. We hope this clarifies the computational positioning and kindly ask the reviewer whether this resolves the remaining concern and whether they would consider increasing their score.

---

### Official Review · Reviewer_tYPo · 2026-03-17

**Soundness:** 2
**Presentation:** 3
**Significance:** 3
**Originality:** 4
**Overall Recommendation:** 5
**Confidence:** 3

**Summary:**

This paper introduces MonoSoup, a data-free, post-hoc weight editing method designed to mitigate representation collapse and OOD degradation during model fine-tuning. MonoSoup achieves comparable benefits using a single checkpoint, reducing storage and computational overhead. The approach applies SVD to each layer's weight update, partitioning it into high-energy directions that capture task-specific adaptations and low-energy directions that preserve robust pre-trained features. By adaptively reweighting these components using an entropy-based effective rank, spectral decay, and geometric alignment, MonoSoup balances specialization and generalization. Empirical evaluations across vision and language models demonstrate that MonoSoup consistently enhances OOD robustness while maintaining strong ID accuracy.

**Compliance With Llm Reviewing Policy:**

Affirmed.

**Final Justification:**

Thank you for the detailed reponse clarifying SVD overhead and OOD robustness. These resolve my concerns regarding efficiency and evaluation rigor. I am raising my score to 5 (Accept).

**Key Questions For Authors:**

1. How does the wall-clock time and memory cost of performing full SVD on every weight matrix scale for larger models like 7B+ parameter LLMs? While avoiding multi-checkpoint training is beneficial, the post-hoc compute required for SVD on massive weight matrices seems non-trivial and is completely missing from the cost analysis.
2. Can the authors justify treating tasks like GSM-Plus and MMLU-Pro-Math as strictly "OOD" relative to GSM8K? These feel more like variations in reasoning difficulty rather than true distribution shifts. Evaluating on completely held-out domains (e.g., code or multilingual tasks) would make the language results as compelling as the vision shifts.
3. How does MonoSoup perform when the fine-tuning task is completely orthogonal to the pre-training distribution (e.g., fine-tuning a natural image model on specialized medical imaging)? It is unclear if the core assumption—that low-energy directions retain valuable pre-training features—holds in such disjoint scenarios.
4. Could the authors clarify the apparent contradiction regarding hyperparameters? Section 4 introduces an elegant, hyperparameter-free method using entropy-based effective rank, yet the Conclusion explicitly cites the reliance on a variance-retention threshold $R$ as a current limitation. Is manual tuning still required in practice?

**Limitations:**

The authors briefly mention the reliance on the threshold $R$ as a limitation in the conclusion, but they fail to adequately address the computational scaling bottlenecks. Performing full SVD on every weight matrix becomes increasingly prohibitive for modern, multi-billion parameter LLMs, and the paper should discuss practical mitigations like randomized or truncated SVD.
The paper also lacks a discussion on potential failure modes. Specifically, if a model is fine-tuned on a dataset with nearly zero mutual information relative to its pre-training data, the preserved "low-energy" directions might inject harmful noise rather than useful residual signals, degrading rather than improving performance.
Finally, there is no discussion of broader societal impacts. While improving OOD robustness is generally positive for model reliability, enabling single-checkpoint robustness with minimal compute could also make it easier for malicious actors to cheaply fine-tune highly robust models for harmful applications.

**Strengths And Weaknesses:**

**Strengths**
1. The empirical results are robust across both vision (CLIP on natural ImageNet shifts) and language domains (Qwen on reasoning benchmarks). The Centered Kernel Alignment analysis provides mechanistic validation.
2. The narrative flow is exceptional. Using the analysis of multi-model geometric alignment as a logical bridge to motivate single-checkpoint spectral decomposition makes the methodology highly intuitive.
3. Fixing representation collapse (the ID-OOD trade-off) using only a single fine-tuned checkpoint is a highly practical contribution. Since model hubs usually only release the single best-performing checkpoint, this makes "soup-like" benefits accessible to everyday practitioners.
4. Mapping the principles of multi-checkpoint ensembling into the internal singular value spectrum of a single model's weight update is a highly novel insight. The derivation of the layer-wise reweighting coefficients—using a combination of spectral decay and cosine similarity to dynamically balance high and low-energy components—is a principled and sophisticated alternative to uniform weight interpolation like Wise-FT.

**Weaknesses**
1. The claim of removing computational overhead is slightly overstated. While it avoids training multiple checkpoints, performing full SVD on every weight matrix of LLMs introduces massive post-hoc compute and memory costs that the authors completely omit from their cost analysis. The definition of OOD for the language modeling experiments (like treating GSM-Plus as OOD relative to GSM8K) is inherently blurrier and less rigorous than the established distribution shifts (ImageNet-A/R/V2) used in the vision domain.
2. The distinction between the automated entropy-based rank method and the fixed threshold ($R$) is slightly muddled in the text. It takes a close reading of the tables and ablation sections to understand exactly when and why the manual $R=0.8$ threshold is still being compared against the "hyperparameter-free" claim.
3. The method assumes that the low-energy directions of the weight update contain useful, generalizable pre-training signals rather than pure noise. This assumption may break down heavily on fine-tuning tasks that are entirely disjoint from the pre-training distribution, a limitation the paper does not explore.

---

> ### Author Rebuttal · Authors · 2026-03-30
>
> We thank the reviewer for their careful and constructive assessment. We are glad they find the empirical results robust across vision and language, the narrative flow exceptional, the single-checkpoint contribution highly practical, and the spectral reweighting insight highly novel. We address each point below.
>
>
> ---
>
>
> **W1 & Q1: SVD computational cost.**
>
>
> We agree the efficiency claim could be better qualified. We have now profiled MonoSoup across four model scales (up to Qwen3-14B). Due to the character limit, we provide the full cost table and discussion in our response to Reviewer nMaW (W2). The key takeaway: MonoSoup takes 2–3 seconds for CLIP models and ~83 seconds for a 14B-parameter model — orders of magnitude cheaper than training the 10+ checkpoints required by Model Soups. We will revise the paper to frame our efficiency claim as relative to multi-checkpoint methods and include the cost analysis in the main text.
>
>
> ---
>
>
> **W2 & Q4: Confusion between entropy-based rank and fixed threshold $R$.**
>
>
> We agree this is confusing as written. The sentence in the Conclusion referring to $R$ as a limitation is a leftover from a previous version and does not reflect the final method. Section 4 introduces $R$ as the natural way to motivate the spectral partitioning, then derives the entropy-based effective rank to remove this hyperparameter entirely. The recommended pipeline is fully hyperparameter-free. Fixed $R$ appears only in ablations to show robustness. We will revise the Conclusion to remove the outdated reference and make the distinction unambiguous throughout the paper.
>
>
> ---
>
>
> **W3 & Q3: Failure mode when fine-tuning and pretraining distributions are disjoint.**
>
>
> We agree with the reviewer's observation and acknowledge this as a limitation. Our experiments cover settings where the pretraining and fine-tuning distributions share substantial structure (CLIP pretrained on web image-text data, fine-tuned on ImageNet; Qwen pretrained on multilingual text, fine-tuned on English math). In such cases, low-energy directions plausibly retain useful pretrained features. When the two distributions are entirely disjoint (e.g., a natural image model fine-tuned on medical imaging), low-energy directions may indeed carry no useful signal, and reweighting could inject noise rather than preserve robustness. We have not tested this regime and will add an explicit discussion of this failure mode in the revision.
>
>
> ---
>
>
> **Q2: OOD evaluation for Qwen — genuinely out-of-distribution tasks.**
>
>
> Due to the character limit, we refer the reviewer to our response to Reviewer nMaW (W1 & Q3), where we present new experiments on two axes of genuine OOD evaluation: cross-lingual (MGSM, 8 languages) and cross-domain (ARC, HellaSwag, MMLU subsets). MonoSoup recovers multilingual degradation in 7/8 languages and improves cross-domain benchmarks by up to +2.8%, providing substantially stronger evidence for OOD robustness in the language setting.
>
> We present brief results here:
>
> *Cross-lingual OOD (MGSM, 8-shot native CoT).* Since the fine-tuned model was trained exclusively on English math data, non-English languages constitute a genuine distributional shift isolating language from difficulty.
>
> | Language | Pretrained | Fine-tuned | MonoSoup | Δ (FT→Mono) |
> |---|---|---|---|---|
> | English | 45.2% | 40.4% | **47.6%** | +7.2% |
> | Spanish | 35.2% | 33.6% | **37.1%** | +3.5% |
> | French | 35.6% | 33.6% | **37.5%** | +3.9% |
> | German | 28.8% | 24.8% | **30.2%** | +5.4% |
> | Chinese | 40.0% | 39.2% | 38.8% | −0.4% |
> | Japanese | 19.2% | 20.8% | **21.1%** | +0.3% |
> | Russian | 33.6% | 31.6% | **34.8%** | +3.2% |
> | Thai | 22.8% | 20.4% | **23.4%** | +3.0% |
> | **Average** | **32.6%** | **30.6%** | **33.8%** | **+3.3%** |
>
> Fine-tuning degrades performance in 6/8 languages (avg. −2.0pp). MonoSoup recovers this degradation in 7/8 languages and surpasses the pretrained baseline by +1.2% on average—simultaneously preserving task-specific adaptation and restoring multilingual features.
>
>
> *Cross-domain OOD (0-shot).* These benchmarks test entirely different capabilities—science reasoning (ARC), commonsense (HellaSwag), humanities and social sciences (MMLU subsets)—that are absent from the math-focused fine-tuning data. Degradation here would indicate representation collapse of general-purpose knowledge.
>
> | Benchmark | Type | Pretrained | Fine-tuned | MonoSoup | Δ (FT→Mono) |
> |---|---|---|---|---|---|
> | GSM8K | ID | 42.08% | 44.81% | **48.87%** | +4.06% |
> | ARC-Challenge | OOD (science) | 31.48% | 32.94% | **33.11%** | +0.17% |
> | HellaSwag | OOD (commonsense) | 37.43% | 37.92% | 37.85% | −0.07% |
> | MMLU Humanities | OOD | 36.66% | 39.77% | **41.91%** | +2.14% |
> | MMLU Social Sci. | OOD | 47.55% | 52.52% | **55.36%** | +2.84% |
>
> MonoSoup improves ID performance (GSM8K: +4.06%) while maintaining or improving all cross-domain benchmarks, with notable gains on MMLU subsets (+2–3%), particularly domains entirely absent from the math fine-tuning data.

---

> > ### Author Rebuttal · Reviewer_tYPo · 2026-04-03
> >
> > Thank you for the detailed response. The authors have addressed most of my concerns regarding computational overhead, hyperparameter-free claims, and OOD evaluation rigor. The profiling confirms efficiency gains over multi-checkpoint methods, and the new cross-domain experiments validate genuine language model robustness. While limitations regarding fully disjoint pre-training and fine-tuning distributions remain, the empirical evidence is compelling.

---

> > > ### Author Response · Authors · 2026-04-06
> > >
> > > We thank the reviewer for the positive reassessment and are glad the concerns are fully resolved.
> > >
> > > ---
> > >
> > > Regarding the remaining note on disjoint distributions: we agree this is an interesting boundary case and will discuss it in the revision. We note, however, that this limitation applies broadly to the field; pretraining data _mixtures_ for foundation models are typically _not released_, making it difficult to guarantee that any evaluation is _truly disjoint_ from pretraining. Our new cross-lingual and cross-domain experiments are designed to maximize the distributional distance within this practical constraint.
> > >
> > > ---
> > >
> > > Finally, we kindly note that the score has not yet been updated to reflect the resolved status; would the reviewer consider adjusting it?

---

### Decision · Program_Chairs · 2026-04-30

**Decision:**

Accept (regular)

**Comment:**

MonoSoup proposes a post-hoc method that decomposes the task vector of a fine-tuned model via SVD to achieve a balance between ID and OOD performance using only a single checkpoint. It removes the practical limitation of conventional Model Soup, which requires multiple checkpoints, and shows competitive performance across a range of architectures.

Three of the four reviewers (scores 5, 5, 4) recommended acceptance after the rebuttal, agreeing that the concerns about SVD overhead and OOD validation had been adequately addressed. The negative review (score 2) noted conceptual overlap with LoRA-Pro; however, LoRA-Pro focuses on training time, whereas MonoSoup is applied post-hoc on top of a fine-tuned model, so the two operate in different application scenarios.

The AC agrees with the reviewers that evaluating MonoSoup's improvements on top of fine-tuned models is an important direction, including the concern raised by the negative reviewer. However, the AC does not view this as a technical flaw of the paper, and considers it insufficient grounds for rejection. Therefore, the AC recommends this paper for weak acceptance.